# Suppressing microtubule detyrosination augments adeno-associated virus 2 endosomal escape and gene delivery

Shefali Tripathi[1,2,3], Shamshul Huda[1,2,3], Joydipta Kar[1,2], Dinesh Chandra[4], Giridhara R. Jayandharan[1,2,3,*] and Nitin Mohan[1,2,*]

## ABSTRACT

Adeno-associated virus (AAV) is a widely used vector for gene delivery, yet the host intracellular trafficking barriers often limit its therapeutic efficacy. Here, we identify microtubule detyrosination – a microtubule post-translational modification – as a key regulator of AAV2 endo-lysosomal processing. Using super-resolution microscopy (SIM/STORM), we show that upon AAV2 endocytosis, the host upregulates microtubule detyrosination via the GSK3β–CLASP2 signaling axis. Single-particle tracking of recombinant virus reveals that detyrosinated microtubules form a physical and functional barrier, restricting AAV2 motility and promoting lysosomal trapping. Restoring microtubule tyrosination via tubulin-tyrosine ligase overexpression or pharmacological inhibition of detyrosination with parthenolide, enhances endosomal escape and perinuclear accumulation of AAV2, translating to improved gene delivery in host cells. Notably, a clinically relevant pro-drug of parthenolide (DMAPT) also displayed a similar trend of enhancing AAV2-driven coagulation factor IX expression in hemophilia B mouse models. Our findings uncover a host mechanism that reshapes the microtubule landscape to restrict AAV2 trafficking and identify microtubule detyrosination as a novel druggable target to improve AAV-based gene therapy.

KEY WORDS: Adeno-associated virus, AAV, Microtubule post-translational modifications, Parthenolide, Endosomal sorting, Endosomal escape, Hemophilia B

## INTRODUCTION

Viral vectors are at the forefront of gene therapy, enabling efficient delivery of genetic material into target cells (Bulcha et al., 2021). Among them, the adeno-associated virus (AAV) has emerged as a powerful tool valued for its diverse serotypes, high tissue tropism and relatively low immunogenicity (Balakrishnan and Jayandharan, 2014; Naso et al., 2017; Selot et al., 2013; Wang et al., 2019). Although AAV holds therapeutic promise for genetic disorders like hemophilia B [caused by clotting factor IX (FIX) deficiency], its efficacy is limited by crucial bottlenecks. At low doses, protein expressions are suboptimal; in contrast, higher doses risk activating immune responses (High and Anguela, 2016; Manno et al., 2006; Miesbach et al., 2022). Current strategies, such as capsid engineering, aim to evade proteasomal degradation and minimize the immune response. For instance, tyrosine-to-phenylalanine substitutions in the VP3 domain or alterations to SUMOylation or Neddylation post-translational modification sites enhance AAV2 capsid stability and transduction efficiency (Gabriel et al., 2013; Mary et al., 2019a,b; Maurya et al., 2019; Mowat et al., 2014; Zhong et al., 2008). However, intracellular trafficking barriers, particularly inefficient endosomal escape and lysosomal degradation of AAV vectors, remain largely unresolved.

Studies have shown that AAV2 vectors enter cells via clathrin-mediated endocytosis, macropinocytosis or clathrin-independent carrier/glycosylphosphotidylinositol-anchored protein (GPI-AP)-enriched compartments (CLIC/GEEC) pathways (Bartlett et al., 2000; Duan et al., 1999; Nonnenmacher and Weber, 2011; Sanlioglu et al., 2000). Post-cellular entry, AAV2 is sorted through the endo-lysosomal compartments. The acidic pH within late endosomes promotes endosomal escape, where the capsid phospholipase A2 (PLA2) activity results in membrane rupture and the release of AAV2 into the cytosol (Ding et al., 2005; Douar et al., 2001; Stahnke et al., 2011). Subsequently, the nuclear localization signal (NLS) on the capsid facilitates AAV nuclear entry and an uncoating of the vector capsid, enabling gene expression (Nicolson and Samulski, 2014). However, post-endosomal escape, most AAV2 particles are ubiquitylated and degraded in the cytoplasm, with only a fraction reaching the nucleus (Berry and Asokan, 2016; Douar et al., 2001; Zhong et al., 2008). Rapid retrograde transport on microtubules might mitigate this inefficiency by enabling endosomal escape near the nucleus, thereby evading cytoplasmic degradation. Supporting this notion, reports show that nocodazole-induced microtubule disruption impairs dynein-mediated AAV2 transport and gene delivery (Kelkar et al., 2006; Xiao and Samulski, 2012; Zhao et al., 2006).

Microtubule post-translational modifications (PTMs) – such as acetylation, tyrosination and detyrosination – give arise to distinct microtubule subtypes within the cell, differentially regulating the activity of motor proteins and organelle trafficking (Geeraert et al., 2010; Janke and Magiera, 2020; Xie et al., 2010). For example, detyrosinated microtubules influence lysosomal distribution and autophagosome–lysosome fusion (Mohan et al., 2019; Verdeny-Vilanova et al., 2017). Viruses like HIV-1 and influenza A modulate microtubule PTMs to facilitate entry or egress in immune and epithelial cells (Husain and Harrod, 2011; Sabo et al., 2013). Building on this evidence, we hypothesized that microtubule PTMs act as host-cell checkpoints governing cytoplasmic trafficking of AAV with a possible impact on infectivity of the vector.

In this study, we investigated the functional role and therapeutic potential of microtubule PTMs in AAV-mediated gene delivery and

[1]Dept of Biological Sciences and Bioengineering, Indian Institute of Technology Kanpur, Kanpur 208016, India. [2]Mehta Family Centre for Engineering in Medicine, Indian Institute of Technology Kanpur, Kanpur 208016, India. [3]Laurus Centre for Gene Therapy, Indian Institute of Technology Kanpur, Kanpur 208016, India. [4]Dept of Hematology, Sanjay Gandhi Post Graduate Institute of Medical Sciences, Lucknow 226014, India.

*Authors for correspondence (nitinm@iitk.ac.in; jayrao@iitk.ac.in)

ⓘ N.M., 0000-0002-5168-1462

rescuing the phenotype of hemophilia. We prioritized the AAV2 serotype for its hepatic tropism (Ponnazhagan et al., 1997) and the established influence of microtubules on its cellular transport (Xiao and Samulski, 2012). Here, we show that AAV2 infection triggers microtubule detyrosination, which restricts viral motility and promotes its lysosomal entrapment. Conversely, suppressing detyrosination – via tubulin-tyrosine ligase overexpression or pharmacological inhibition with parthenolide – improves AAV2 retrograde trafficking and significantly enhances transduction efficiency in hepatic cells (Huh7). Notably, the clinically relevant parthenolide pro-drug DMAPT recapitulates these effects in hemophilia B mice, boosting FIX expression. Our findings establish microtubule detyrosination as a crucial barrier to AAV2 gene delivery, offering a host-directed and pharmacologic strategy to improve gene therapy outcomes.

## RESULTS
### AAV2 endocytosis induces microtubule detyrosination via GSK3β-CLASP2 signaling

We investigated microtubule-PTM remodeling during AAV2 entry using a pulse-chase approach (see Materials and Methods), where Huh7 cells exposed to AAV2 [$2\times10^4$ vector genomes (vgs)/cell] were imaged with super-resolution structured illumination microscopy (SIM) at definite time points post-endocytosis (Fig. 1A–C). Endocytic AAV2 puncta increased progressively up to 4 h post entry, followed by a decline by 8 h – consistent with cytoplasmic degradation or nuclear translocation (Fig. 1A). Strikingly, detyrosinated microtubule levels mirrored this trend, with an increase at 4 h. We measured detyrosinated microtubule levels relative to tyrosinated microtubules and found that they peaked at 4 h (0.35±0.02; mean±s.d.) and returned to baseline by 8 h (0.14±0.03) (Fig. S1A; Fig. 1B). We confirmed the antibody specificity to detyrosinated microtubule by titration (see Materials and Methods) and confocal imaging of detyrosinated microtubules (Fig. S1B) and further validated it by showing loss of signal in cells upon vasohibin 2 (VASH2; a tubulin detyrosination enzyme) knockdown (Fig. S1C). Furthermore, whereas acetylated microtubules were undetectable (Li et al., 2020; Lu et al., 2007) (Fig. S1D) with AAV2 endocytosis, we observed a slightly increased microtubule polyglutamylation (poly E) (Fig. S1E).

To further confirm our observations, we assessed microtubule PTM levels in AAV2-infected cells (4 h) in comparison to mock-treated control by immunoblotting. AAV2-treatment caused a 3.5-fold increase in detyrosinated microtubule and a slight decline in tyrosination, consistent with SIM imaging. We did not observe a significant change in acetylation, whereas microtubule polyglutamylation increased 1.7-fold (Fig. 1E,F). Given that it was challenging to observe PolyE microtubules in Huh7 cells (Fig. S1E), we imaged these modifications in BS-C-1 cells to test whether they overlap with detyrosinated microtubules. Consistent with previous findings (Ebberink et al., 2023), our confocal imaging and intensity profile analysis confirmed the colocalization of polyglutamylation and detyrosination (Fig. S1F,G). Given that AAV2-induced changes in microtubule modification were highest for detyrosination, we proceeded with further dynamic analyses focusing on microtubule detyrosination.

This dynamic was conserved in BS-C-1 cells, which exhibit higher baseline detyrosination (0.25±0.04, mean±s.d.). Here, AAV2 endocytosis transiently increased detyrosination with peak levels observed 4 h (0.6±0.07), followed by a decline to baseline by 8 h (Fig. S1H,I). Notably, detyrosination persisted in 75.7±9% (mean±s.d.) of Huh7 cells at 4 h but declined to 54.8±11.2% and 29.1±3.6% by 8 h and 24 h, respectively (Fig. S1J,K).

Next, we investigated how AAV2 entry increases microtubule detyrosination. Several co-receptors involved in AAV2 endocytosis are receptor tyrosine kinases (RTKs) (Pillay and Carette, 2017), suggesting that AAV2 triggers an RTK signaling cascade leading to detyrosination. RTK activation typically stimulates AKT family protein phosphorylation downstream (Schlessinger, 2000). Specifically, we investigated the RTK–AKT downstream kinase, glycogen synthase kinase 3β (GSK3β) (Cross et al., 1995), which has been shown to suppress the microtubule-stabilizing protein CLASP2 (Kumar et al., 2009; Pemble et al., 2017). Microtubule detyrosination mediated by the carboxypeptidase VASH1/2, is correlated with microtubule stability (Aillaud et al., 2017; Li et al., 2019). We thus propose that AAV2-triggered RTK activation leads to AKT-mediated phosphorylation of GSK3β, relieving its repression on CLASP2, thereby promoting CLASP2 microtubule binding, which likely facilitates detyrosination by VASH1/2 (Fig. 1G).

Supporting this model, AAV2 infection, markedly increased phospho-GSK3β levels compared to mock controls in Huh7 cells (Fig. 1H–J). Furthermore, a constitutively active, phosphorylation-resistant GSK3β mutant (S9A) blocked AAV2-induced detyrosination, indicating that rather than acting through a direct mechanism, AKT promotes detyrosination indirectly by inhibiting GSK3β (Fig. 1K,L). Notably, the same results were seen in BSC-1 cells, suggesting that this pathway is conserved (Fig. S1L). We next assessed CLASP2, which showed an increased microtubule association following AAV2 endocytosis compared to mock controls (Fig. 1M,N). Furthermore, whereas overexpression of wild-type CLASP2 induced detyrosination only in the AAV2-treated cells, with CLASP2 showing colocalization with detyrosinated tracks (Fig. 1O,P), a GSK3β-insensitive CLASP2 mutant (9xS/A) was sufficient to promote detyrosination even without AAV2 (Fig. 1O, CLASP2 9xS/A; Fig. 1P). These observations indicate that merely overexpressing wild-type CLASP2 does not cause detyrosination. In contrast, overexpression of a GSK3β-insensitive mutant of CLASP2 does induce detyrosination. This suggests that GSK3β might inhibit CLASP2 binding to microtubules. When treated with AAV2, RTK signaling is activated, leading to the inhibition of GSK3β and allowing CLASP2 to associate with microtubules. This binding of CLASP2 could play a role in promoting microtubule detyrosination through mechanisms that are still to be investigated. This mechanism also supports the host–pathogen interaction driven pathway as reported previously that HIV induces microtubule stabilization by exploiting the CLASP2 (Mitra et al., 2020). Furthermore, GSK3β might also regulate VASH1/2-induced detyrosination through CLASP2-independent mechanisms, which warrants further investigation.

### AAV2 accumulates on detyrosinated microtubules, which reduces its motility

As previous results indicate that AAV2 infection promotes microtubule detyrosination via host signaling pathways, we next asked whether this post-translational modification affects the spatial distribution and motility of AAV2. Furthermore, with the help of SIM imaging and quantification (Fig. S2A), we observed a time-dependent accumulation of AAV2 particles on detyrosinated microtubules in both Huh7 (Fig. 2A–C) and BS-C-1 cells (Fig. S2B–D). Super-resolution STORM imaging further confirmed the localization of AAV2 on detyrosinated microtubules with ~20 nm precision (Fig. S2E,F). These findings demonstrate that AAV2 endocytosis induces a ~25% increase in detyrosinated microtubules, leading to the accumulation of ~35% of AAV2 and an ~1.4-fold enrichment on detyrosinated microtubules at 4 h (Fig. 2C),

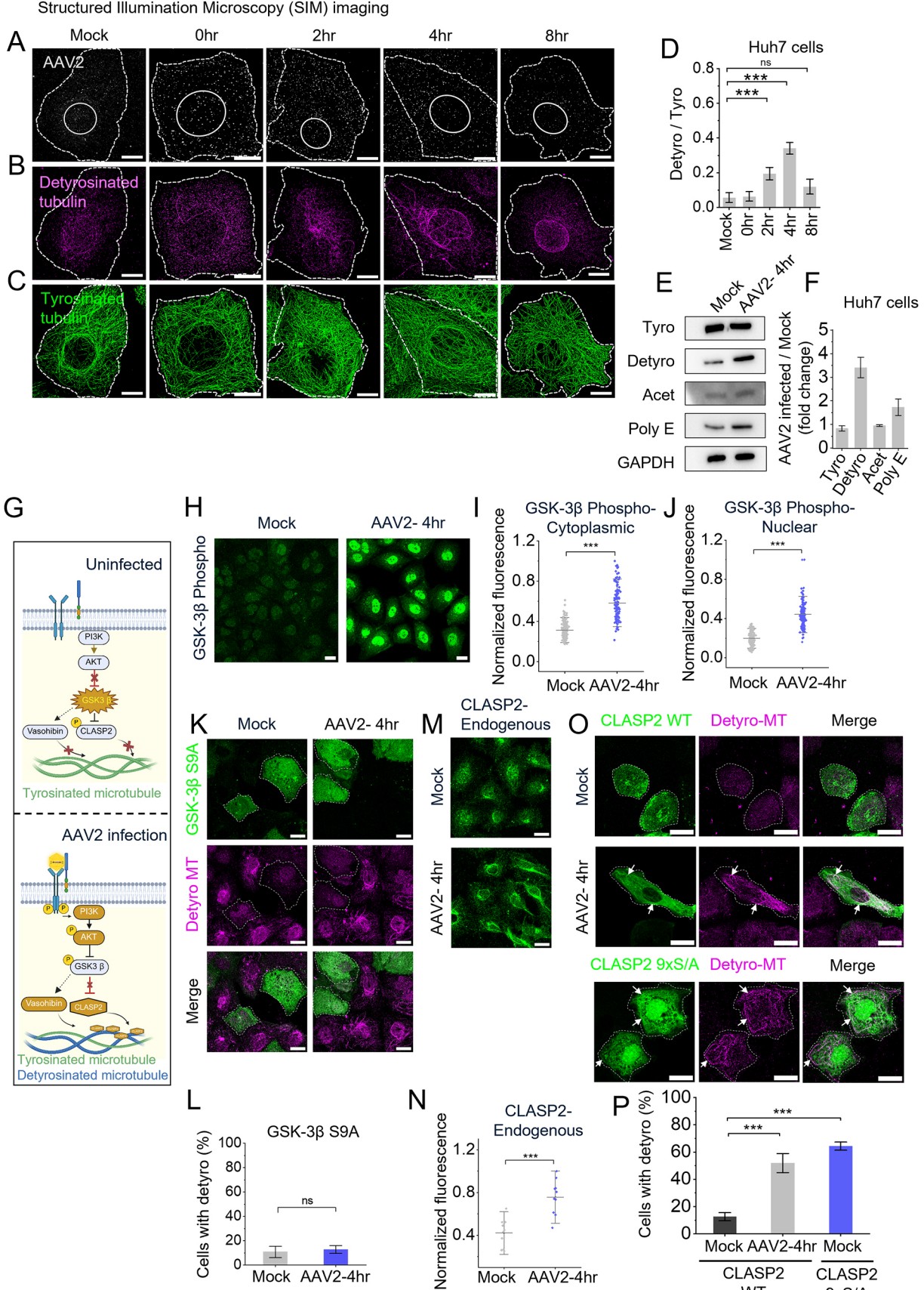

**Fig. 1.** See next page for legend.

**Fig. 1. AAV2 endocytosis upregulates microtubule detyrosination via GSK3β-CLASP2 signaling.** (A–C) Three-color SIM imaging of AAV2 (A), detyrosinated microtubules (B) and tyrosinated microtubules (C) in Huh7 cells at 0, 2, 4 and 8 h post-AAV2 treatment. At 8 h infection, only a small fraction of detyrosinated microtubules remain around the nucleus. Dashed lines highlight cell edge; solid lines highlight location of nucleus. (D) Quantification represents the change in ratio of detyrosinated to tyrosinated microtubules in Huh7 across time points post-AAV2 treatment (n=15 cells analyzed across three independent experiments). (E,F) Western blot representing the blot intensities of microtubule-PTMs and GAPDH in mock (E). Qualification of respective blot intensities representing the fold change in microtubule PTMs (n=3, three independent replicates) (F). (G) Schematic model of AAV2-induced signaling via RTK leading to GSK3β inhibition and CLASP2-mediated microtubule detyrosination. Created in BioRender by Jayandharan, G. R., 2025. https://BioRender.com/85wkr7t. This figure was sublicensed under CC-BY 4.0 terms. (H–J) Immunostaining (H) and quantification of cytoplasmic (I) and nuclear (J) phospho-GSK3β (Ser9) in mock- and AAV2-infected Huh7 cells at 4 h, showing increased Ser9 phosphorylation (n=125 cells, analyzed across three independent experiments). (K–L) Expression of the constitutively active GSK3β S9A mutant (green) and quantification of cells with detyrosinated microtubules (magenta) under mock or AAV2-infected conditions (n=190 cells, analyzed across three independent experiments). (M,N) Immunostaining (M) and quantification (N) of endogenous CLASP2, showing increased microtubule association in infected cells (in triplicate). Arrows indicate CLASP2 localized to microtubules. (O,P) Modulation of detyrosinated microtubules in cells overexpressing CLASP2 WT (green) or CLASP2 mutant-9xS/A (green) with AAV2 treatment compared to mock controls (n=160 cells, analyzed across three independent experiments). Arrows highlight the CLASP2 colocalizing with detyrosinated microtubules. In all graphs, results are shown as mean±s.d. ***P<0.001; ns, not significant (unpaired two-tailed t-test). Scale bars: 10 μm (A–C); 20 μm (H,K,M,O).

suggesting that detyrosination acts as a roadblock to AAV2 retrograde transport. Furthermore, the subsequent decline in cytoplasmic AAV2 at 8 h indicates that detyrosination likely promotes AAV2 degradation.

To resolve whether detyrosination impedes AAV2 retrograde transport, we used the correlative live-cell and STORM imaging approach (Bálint et al., 2013; Mohan et al., 2019; Verdeny-Vilanova et al., 2017). At 1-h post-endocytosis, we tracked Cy3-tagged AAV2 in live cells for ∼1 min, followed by in situ immunostaining and STORM imaging of microtubules within the same cells. To assess the AAV2 dynamics precisely, we mapped the single-particle trajectories (SPTs) trajectories on detyrosinated or tyrosinated microtubules. By mapping AAV2 SPTs onto the STORM image (Fig. 2D,E; Fig. S2G) and segmenting processive (run) and non-processive (pause) states (Fig. 2G,G′; Fig. S2G), we quantified AAV2 motility, using a custom SPT analysis algorithm (Bálint et al., 2013). Our results reveal that AAV2 transitions from run to long-pause states (Fig. 2F–G′; Movie 1) or remains static (confined motion) on detyrosinated microtubules (Fig. S2H; Movie 2). In contrast, its movement is not impeded in the subcellular space devoid of detyrosinated microtubules (Fig. S2I; Movie 3) or when moving on tyrosinated microtubules (Fig. S2G,J; Movies 4 and 5).

Quantitative analysis demonstrated that detyrosinated microtubules reduced displacement and speed of AAV2 by 60% (mean= 0.32±0.1 μm/s versus 0.8±0.2 μm/s on tyrosinated tracks, mean±s.d.; Fig. 2H) and shortened run lengths by ∼78% (0.75±0.21 μm versus 2.16±0.55 μm; Fig. 2I). Detyrosination also increased the pause frequency (mean=4.85±0.9 pauses/track versus 1.7±0.32 pauses/ track), Fig. 2J) and prolonged pause durations (mean=3.04±0.6 s versus 1.1±0.2 s, Fig. 2K). Together, these data demonstrate that detyrosination acts as a bio-physical barrier to AAV2 motility, either by trapping it or intermittently obstructing its movement, potentially limiting its retrograde transport and nuclear entry.

## Suppressing microtubule detyrosination accelerates AAV2 transport towards the nucleus

Next, we modulated detyrosination and tyrosination levels to assess their impact on AAV2 motility and cytoplasmic distribution. Whereas GFP vector-only overexpression does not modulate the basal level of detyrosinated microtubule in Huh7 cells (Fig. S3A), the overexpression of the tubulin-detyrosination enzyme VASH1/2 (Aillaud et al., 2017; Ramirez-Rios et al., 2023; Tang et al., 2023) increased detyrosinated microtubules in Huh7 cells (Fig. S3B) and reduced AAV2 motility, whereas overexpression of the tyrosination enzyme tubulin tyrosine ligase (TTL) (Pietsch et al., 2024; Tang et al., 2023) increased the ratio of tyrosinated to detyrosinated microtubules (Fig. S3B) and enhanced AAV2 displacement compared to untreated controls or GFP vector control (Fig. 3A,B; Movies 6, 7 and 8; Fig. S3C, GFP vector control). We detected detyrosinated microtubules even with TTL overexpression (Fig. S3D), likely because TTL preferentially acts on detyrosinated tubulin dimers but not polymerized detyrosinated microtubules (Prota et al., 2013). Nonetheless, TTL overexpression significantly increased overall levels of α-tubulin and tyrosinated microtubules (Fig. S3E,F), correlating with enhanced AAV2 motility. These findings suggest that both increasing tyrosination and reducing detyrosination can promote AAV2 transport.

Additionally, parthenolide, a well-known VASH1/2 inhibitor (Fonrose et al., 2007; Gobrech et al., 2024), completely suppressed microtubule detyrosination (at 20 μM, see Materials and Methods; Fig. S3G,H) and significantly increased AAV2 motility (Fig. 3A,B, Movie 9). Additionally, AAV2 exhibited higher speed and run length in parthenolide-treated cells in comparison to the untreated control cells (Fig. S3I–K).

To examine the spatial distribution of AAV2 under variable tyrosination and detyrosination levels, we divided cells into three zones – inner (perinuclear), middle (mid-cytoplasmic) and outer (near the plasma membrane) (see Materials and Methods, Fig. 3D). We then quantified AAV2 localization in these zones over time. VASH2 overexpression restricted most AAV2 to the outer zone, with only a fraction (Fig. 3C,E; mean=28.25±5% at 4 h) reaching the perinuclear zone. In contrast, TTL overexpression facilitated the accumulation of a larger pool (mean=75±0.6% at 4 h) of AAV2 near the nucleus in comparison to the untreated (Fig. 3C,E) or GFP vector controls with AAV2 infection (Fig. S3L). Similarly, parthenolide-mediated suppression of detyrosination significantly increased AAV2 accumulation in the inner zone (mean=69.6±7% at 4 h). These findings indicate that detyrosinated microtubules impede AAV2 retrograde transport to the nucleus, whereas suppressing detyrosination or enhancing tyrosination accelerates perinuclear accumulation, potentially improving AAV2 nuclear entry.

## Microtubule tyrosination enhances AAV2 endosomal escape

To determine how the tyrosination or detyrosination status of microtubules regulates AAV2 endosomal sorting, we tracked the colocalization of AAV2 with Rab5 (early endosomes), Rab7 (late endosomes) and LAMP1 (lysosomes) in parthenolide-treated cells over time in comparison to untreated cells. Parthenolide treatment significantly reduced AAV2 retention in early endosomes (Fig. S4A,B), indicating a rapid sorting of AAV2 from early to late endosomes. Notably, unlike the progressive accumulation of AAV2 in late endosomes and lysosomes observed in control cells, parthenolide treatment accelerated AAV2 exit from late endosomes (Fig. 4A,B) and reversed its accumulation in lysosomes (Fig. 4C,D), suggests that microtubule tyrosination enhances endosomal escape of AAV2 and diverted it from lysosomes.

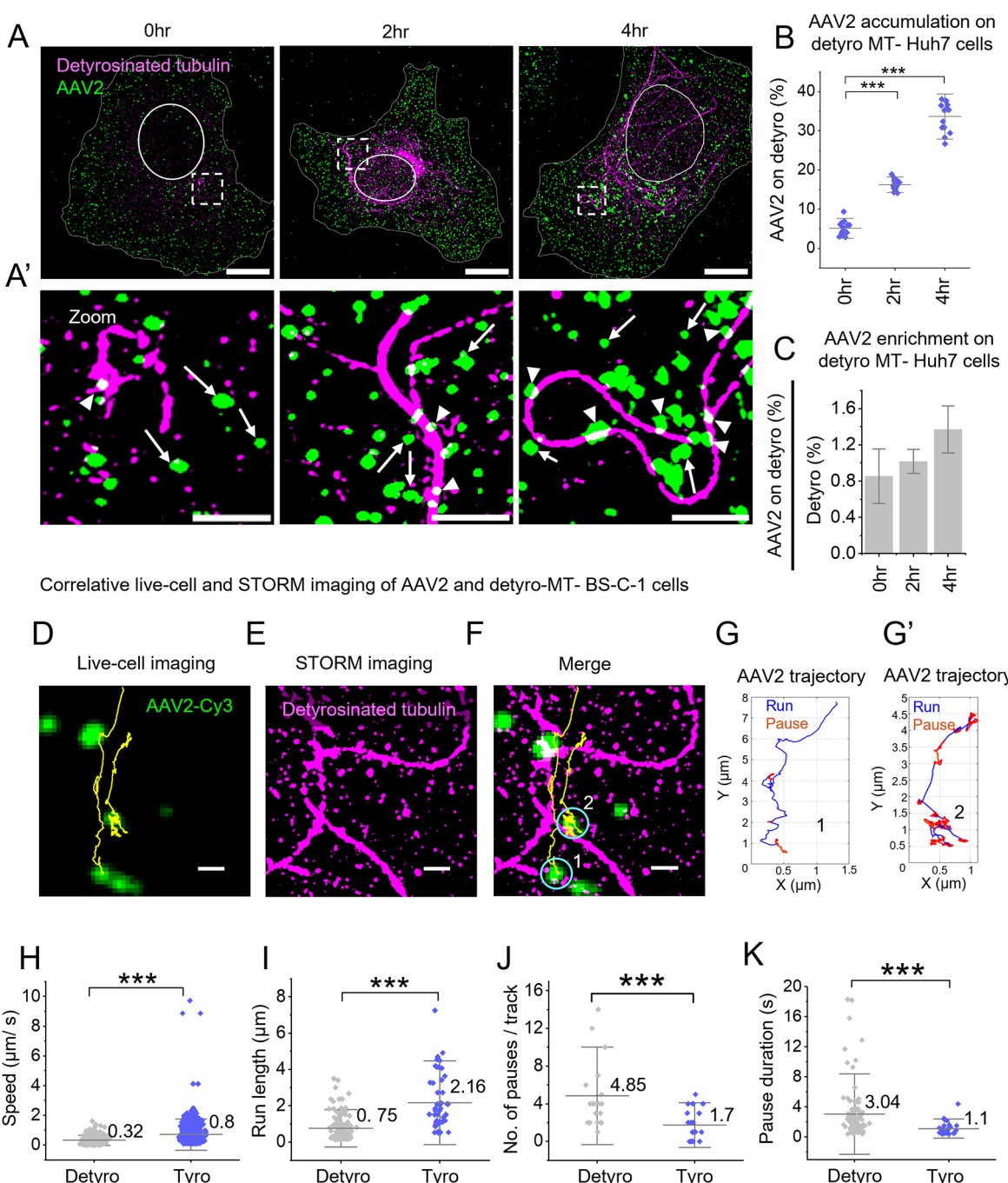

**Fig. 2. Detyrosinated microtubules influence AAV2 distribution and motility dynamics.** (A,A′) Two-color SIM images of Huh7 cells treated with AAV2 (green) and immunostained for detyrosinated microtubules (magenta) at 0, 2 h and 4 h. Solid lines highlight location of nucleus. (A′) Magnified view of the respective areas marked in A. Arrowhead and arrows show AAV2 localized to and not localized to detyrosinated microtubules, respectively. (B,C) Quantification of the localization (B) and enrichment (C) of AAV2 on detyrosinated microtubules at different time points post-infection in Huh7 cells (10 cells analyzed for each time point; *n*=10). (D–F) Correlative live-cell and STORM imaging of AAV2 and detyrosinated microtubules in BS-C-1 cells. Trajectory of AAV2 motility (yellow) from the live-cell imaging (D). STORM image of the detyrosinated microtubules (magenta) from the same region (E). AAV2 trajectory from A overlaid with STORM image of detyrosinated microtubule from E (F). (G,G′) Color-coded AAV2 trajectory shows processive run phases in blue and non-processive pause phases in red from AAV2 particle 1 (G) and AAV2 particle 2 (G′). (H–K) Quantitative analysis of speed (H), run length (I), number of pauses per track (J) and fraction of time spent pausing (pause duration) (K) of AAV2 motility on detyrosinated microtubule compared to tyrosinated microtubule (20 tracks analyzed per condition: *n*=20). In all graphs, results are shown as mean±s.d. ***P<0.001 (unpaired two-tailed *t*-test). Scale bars: 10 μm (A); 2 μm (A′); 1 μm (D–F).

To directly assess endosomal escape facilitated by tyrosinated microtubules, we employed the Calcein Green assay (see Materials and Methods). Calcein, a membrane-impermeable dye, enters cells via endocytosis and its punctate fluorescence (confined to intact endosomes) transitions to a diffuse cytoplasmic signal upon AAV2-induced membrane rupture (Fig. 4E). In cells overexpressing VASH2

Journal of Cell Science

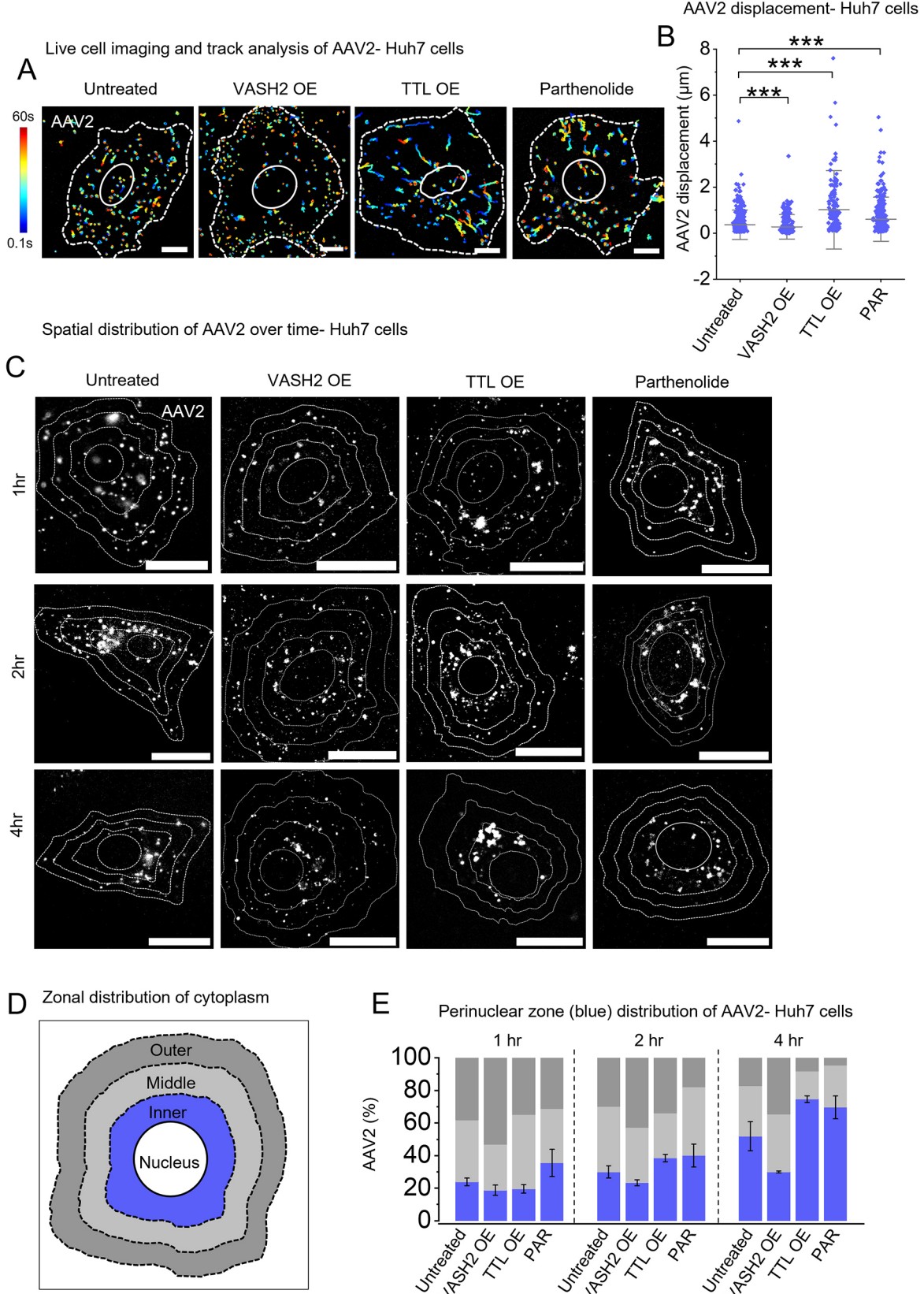

**Fig. 3.** See next page for legend.

(promoting detyrosination), Calcein remained punctate until 8 h post-entry, indicating minimal escape (Fig. 4F,G). Conversely, TTL overexpression (inducing tyrosination) or parthenolide treatment triggered Calcein cytoplasmic diffusion within 1 h, with signal intensity increasing over time, demonstrating robust AAV2 escape (Fig. 4F,G). These findings suggest that suppressing detyrosinated

**Fig. 3. Suppression of detyrosinated microtubules improves AAV2 motility and perinuclear accumulation.** (A,B) Live-cell imaging and track displacement measurement of Cy3-tagged AAV2 in Huh7 cells in untreated control conditions, in cells overexpressing (OE) VASH2–GFP or TTL–GFP or in cells pretreated with parthenolide. Color-coded trajectories indicate blue for the start and red for the end of the tracks acquired for 60 s duration. Dashed lines highlight cell edge; solid lines highlight location of nucleus (A). Quantification of AAV2 track displacement obtained from movies in A (B). Results show mean±s.d. for ~500 trajectories from $n$=4 cells analyzed for each condition from three replicates. (C–E) Time-dependent spatial distribution analysis of AAV2 in Huh7cells. TIRF imaging of AAV2-Cy3 (white) in cells fixed at 1 h, 2 h and 4 h after AAV2 endocytosis (C). Schematic representation of the division of the cytoplasmic region into three equal zones: outer (near plasma membrane), inner (perinuclear) and middle (between zone 1 and 2), to analyze the spatial distribution of AAV2 within cells (D). Percentage of AAV2 distribution across the zones over time in Huh7 cells (E). Bars represent the mean±s.d. for $n$=10 cells analyzed in each condition from three replicates. ***$P$<0.001 (unpaired two-tailed $t$-test). Scale bars: 10 µm (A); 20 µm (C).

microtubules through TTL overexpression or parthenolide treatment prevents lysosomal entrapment and promotes AAV2 endosomal escape, likely facilitating its nuclear translocation.

## Suppression of microtubule detyrosination enhances AAV2 transduction efficiency

Although endosomal escape is essential for AAV2 nuclear entry, it exposes the virus to ubiquitylation and proteasomal degradation. To determine whether suppressing microtubule detyrosination preferentially promotes AAV2 nuclear entry over degradation, we quantified transduction using an AAV2 vector with a GFP reporter transgene. Overexpression of VASH2 (inducing detyrosination) reduced GFP-positive cells to 14.6±2.8% (Fig. 5A,B). To confirm that this effect was specifically due to the catalytic activity of VASH2, we compared it with cells expressing the catalytically inactive mutant VASH2 C158A (Aillaud et al., 2017). In this case, 54±3.5% (mean±s.d.) of cells were transduced with AAV2–EGFP, similar to the untreated AAV2 transduced condition in Huh7 cells (Fig. S5A), indicating no preference or no reduction in transduction efficiency. Using shRNA – thereby suppressing microtubule detyrosination in Huh7 and BS-C-1 cells compared to scramble controls (Fig. S5B and S1C, respectively) – led to an ~75% increase in AAV2–GFP transduction. These results together confirm that the observed effect on AAV2 transduction is dependent on the enzymatically active form of VASH2. To confirm this was not restricted to VASH2, we also tested with VASH1 knockdown, which similarly enhanced transduction in Huh7 cells (Fig. S5B,C). For greater accuracy, we examined both VASH1 and VASH2 (hereafter VASH1/2) single knockdowns in BS-C-1 cells, which showed ~23% endogenous detyrosination (Fig. 1E; Fig. S1F). Consistent with this, VASH1/2 shRNA increased AAV2 transduction, confirming that the enhancement was directly due to suppression of detyrosinated microtubules (Fig. S5D,E). We then observed that the TTL overexpression (promoting tyrosination) markedly increased transduction to 82.3±1.5% (Fig. 5A,B). AAV2–EGFP expression inversely correlated with VASH2 levels but increased dose dependently with TTL (Fig. S5F,G), confirming that detyrosination impedes AAV2 nuclear translocation.

Pharmacological suppression of detyrosination with parthenolide further supported these findings. Parthenolide-pretreated cells exhibited a ~2-fold increase in GFP fluorescence intensity and an ~1.5-fold higher percentage of GFP-positive Huh7 cells compared to untreated controls, across a range of vector doses ($5\times10^3$ or $2\times10^4$ vgs/cell; Fig. 5C,D). Consistently, flow cytometry analysis

revealed 3.6-fold more GFP-positive Huh7 cells (mean=29.4±0.91% vs 8±0.16% in controls; $P\leq2.4\times10^{-6}$) and ~1.5-fold higher GFP intensity following parthenolide treatment (Fig. 5E–G), demonstrating enhanced AAV2 transduction with pharmacological detyrosination suppression *in vitro*.

To validate the therapeutic relevance of suppressing detyrosination as a strategy to improve AAV2-mediated gene therapy *in vivo*, we used a hemophilia B mouse model (Fig. 5H, see Materials and Methods). We used the pro-drug dimethylaminoparthenolide (DMAPT) due to its bioavailability and prior use in suppressing detyrosinated microtubule in animal models (Leibinger et al., 2023). Mice receiving scAAV2-LP1-h.FIX vector ($5\times10^{10}$ vgs/mouse) pretreated with DMAPT showed an elevated FIX coagulant activity (mean=54.47±22.9%, Fig. 5I) compared to mice without DMAPT pretreatment (mean=35.84±24.33%, Fig. 5I). Additionally, in liver tissues of DMAPT-treated mice, FIX expression increased (Fig. 5J). Immunohistochemistry reveals the reduced microtubule detyrosination over tyrosination (Fig. 5K–N) further supporting the role of detyrosination suppression in enhancing AAV2 efficacy.

However, the enhanced FIX activity observed in the mouse model cannot solely be attributed to detyrosination suppression given the known pleiotropic effects of parthenolide (Carlisi et al., 2016; Eibes et al., 2025; Hehner et al., 1999; Hotta et al., 2021; Liu et al., 2009). Additional studies are required to validate the drug and vector dosage, as well as observed phenotypic effect during suppression of microtubule detyrosination. Nonetheless, our *in vitro* and *in vivo* results together demonstrate that suppressing microtubule detyrosination – a host-cell barrier that traps AAV2 on retrograde transport pathways – enhances viral nuclear entry and gene delivery efficacy, offering a clinically actionable strategy to optimize AAV-based therapies.

## DISCUSSION

Our study uncovers a new role for detyrosinated microtubules in cellular defense against AAV2 infection in liver cells and tissues. Upon the AAV2 infection, the host cell increases the detyrosinated microtubule levels to impede AAV2 trafficking towards the nucleus, limiting its transduction. Suppressing microtubule detyrosination enhances AAV2-mediated factor IX gene transduction efficiency in liver cells and hemophilia B mouse models, offering a promising strategy for augmenting AAV-mediated gene therapy.

Viruses often hijack host cytoskeletal pathways to facilitate their intracellular transport. For example, HIV-1 promotes the formation of stable microtubules (Sabo et al., 2013) via the diaphanous-related formin (DRF) proteins Dia1 and Dia2 (Delaney et al., 2017), whereas microtubule-associated proteins (MAPs), such as CLASP2 support its intracellular motility (Mitra et al., 2020). Intriguingly, Wnt/β-catenin-GSK3β signaling is reported to modulate HIV replication (Wen et al., 2022). Similarly, influenza A virus infection increases microtubule acetylation (Husain and Harrod, 2011), and Tubacin mediated suppression of acetylation reduces viral replication (Husain and Cheung, 2014). Notably, RTK/AKT pathways are activated during influenza A infection as well, promoting anti-apoptotic responses (Ehrhardt et al., 2007), and likely facilitating cytoskeletal remodeling for viral trafficking.

Our data reveal that AAV2 infection engages a related but distinct pathway. Specifically, AAV2 internalization activates RTK–AKT–GSK3β signaling, which in turn elevates detyrosinated microtubule levels. Notably, we show that this detyrosination is not a direct effect of viral components but rather a downstream consequence of host signaling responses triggered by AAV2 entry. Unlike HIV-1 or influenza A, where stable acetylated

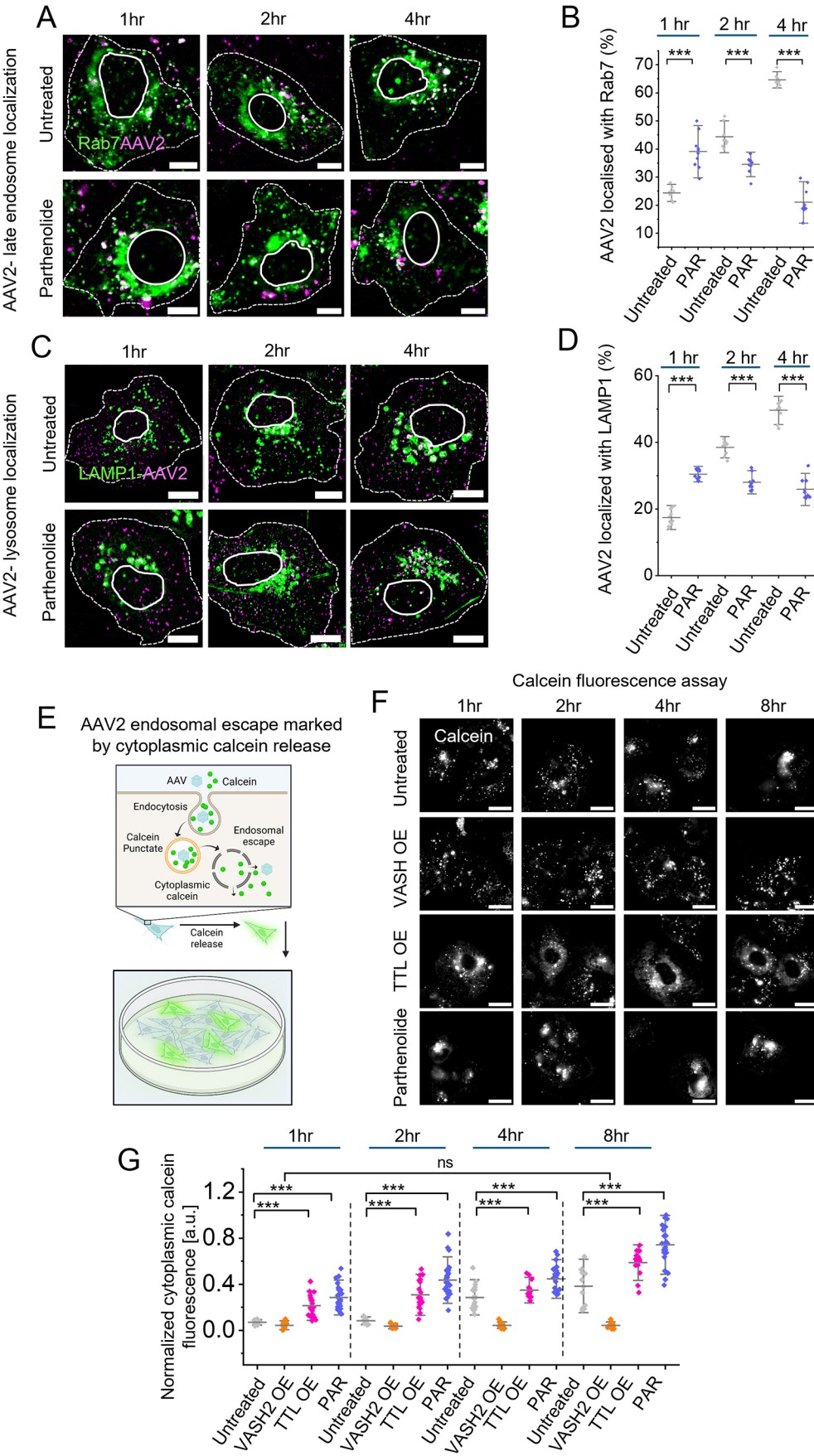

**Fig. 4. Tyrosinated microtubules promote AAV2 endosomal escape.** (A,B) Time-dependent colocalization analysis of AAV2 with late endosomes in Huh7 cells. Confocal imaging of Rab7–GFP (green) along with Cy3-tagged AAV2 (magenta) at different time points in Huh7 cells in untread conditions and pre-treated with parthenolide (A). Percentage of AAV2 colocalized with Rab7 decorated late endosomes over time analyzed from ten images (*n*=10) of three replicates (B). (C,D) Confocal imaging (C) and quantification (D) of AAV2 (magenta) colocalization with lysosomes decorated by LAMP1 (green) (C). Whereas untreated control cells show significant accumulation of AAV2 in lysosomes over time, parthenolide-treated cells show no significant change in lysosomal colocalization with time analyzed from ten images (*n*=10) of three replicates (D). In A and C, dashed lines highlight cell edge; solid lines highlight location of nucleus. (E–G) AAV2 endosomal escape measurement using Calcien fluorescence assay in Huh7 cells. Schematic representation of Calcien fluorescence assay. Endosomal Calcein fluorescence appears as a punctate signal whereas, AAV2 mediated endosomal rupture leads to diffused cytoplasmic Calcein fluorescence (E). Created in BioRender by Jayandharan, G. R., 2025. https://BioRender.com/te8cvym. This figure was sublicensed under CC-BY 4.0 terms. Live-cell imaging of Calcein (white) in AAV2 infected cells at 1, 2, 4 and 8 h post AAV2 endocytosis in untreated control, VASH2–GFP overexpression (OE), TTL–GFP overexpression, or parthenolide pretreated conditions (F). Quantification of cytoplasmic Calcein fluorescence over time (*n*=22 from three replicates) (G). Bars represent the mean±s.d. a.u., arbitrary units. ***P<0.001; ns, not significant (unpaired two-tailed *t*-test). Scale bars: 10 µm (A,C); 20 µm (F).

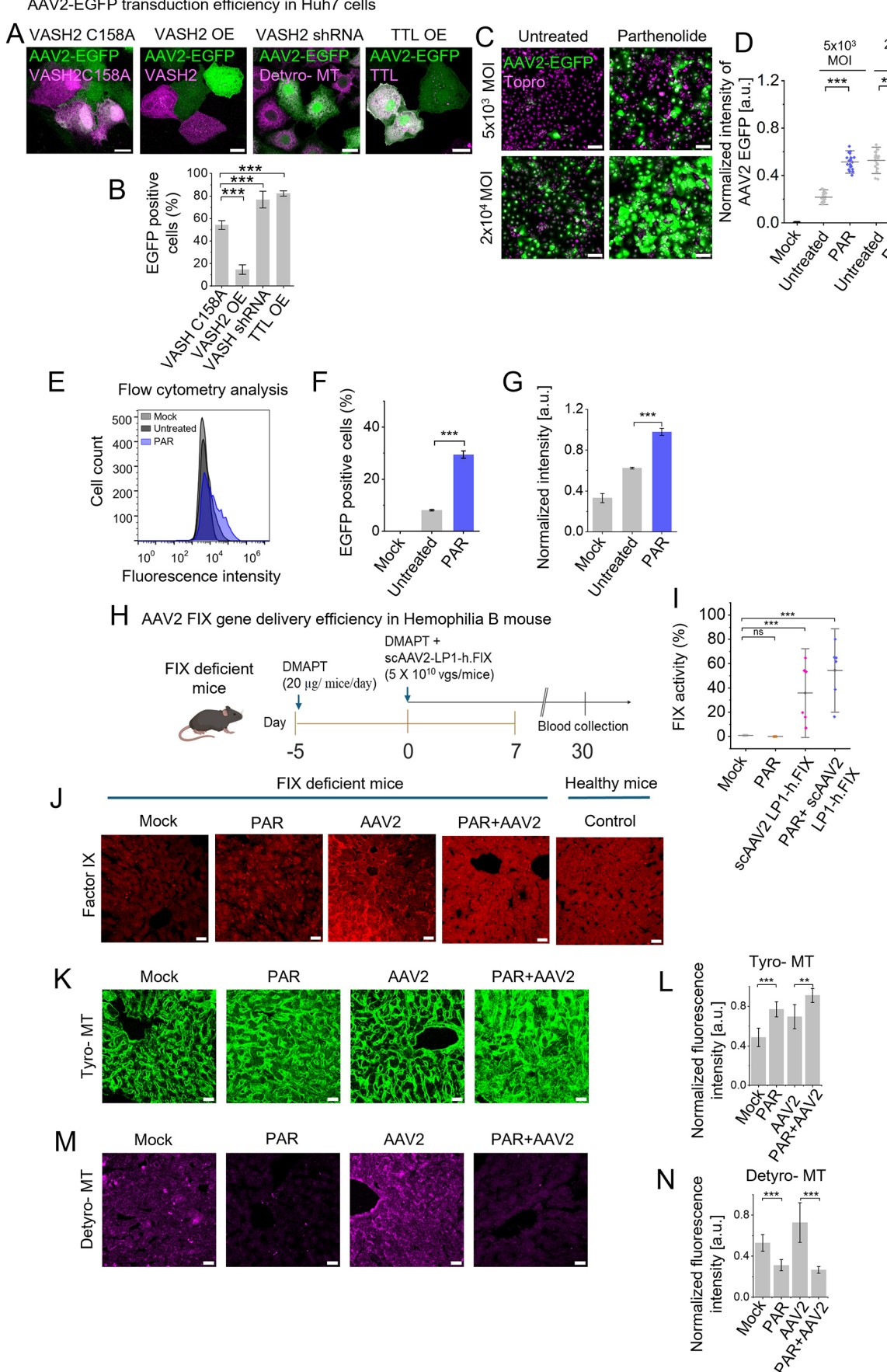

**Fig. 5.** See next page for legend.

**Fig. 5. Detyrosinated microtubule modulation and combinatorial parthenolide plus AAV2-FIX therapy in cellular and animal models.** (A,B) Assessment of the impact of VASH2 and TTL overexpression and VASH2 shRNA on AAV2 transduction efficiency. Confocal imaging of Huh7 cells expressing VASH2 C158A (dead mutant), overexpressing (OE) VASH2, expressing VASH2 shRNA or with TTL overexpression (magenta) infected with AAV2–EGFP, where expression of the EGFP transgene (green) indicates AAV2 transduction (A). Percentage of EGFP-positive cells among VASH2 C158A, VASH2 overexpression, VASH2 shRNA or TTL-overexpressing populations, highlighting their influence on transduction. $n=100$ cells analyzed for each condition from triplicate experiments (B). (C,D) Huh7 cells, untreated or pretreated with parthenolide, were infected with AAV2–EGFP at low (MOI: $5×10^3$ vg/cell) and high (MOI: $2×10^4$ vg/cell) vector doses. Confocal imaging shows GFP expression (green) in infected cells, with TO-PRO staining nuclei (magenta) (C). Fluorescence intensity quantification confirmed enhanced AAV2–EGFP expression in parthenolide-treated cells at both MOIs ($n=18$ images from triplicate experiments) (D). (E–G) Flow cytometry analysis (E) and quantification of EGFP-positive cells (F) and normalized fluorescence intensity (MOI: $5×10^3$ vg/cell) (G), also indicate improved transduction efficiency with parthenolide treatment. $n=10,000$ cells were analyzed by flow cytometry for each condition for triplicate experiments. (H–J) Schematic illustrates the *in vivo* therapeutic evaluation of parthenolide combined with scAAV2-LP1-h.FIX in a hemophilia B mouse model. Created in BioRender by Jayandharan, G. R., 2025. https://BioRender.com/8105pml. This figure was sublicensed under CC-BY 4.0 terms (H). Parthenolide was administered from day −5 to day −1, followed by AAV2-FIX administration on day 1 and blood collection on day 30 for FIX activity analysis ($n=5$–7 mice/group;). A chromogenic assay (I) compared FIX activity across four groups: mock (untreated hemophilia B mice), PAR (parthenolide only), AAV2 (vector without drug), and PAR+AAV2 (combination therapy). Immunohistochemistry and confocal imaging of liver sections showed FIX expressions (red) in the PAR+AAV2 group compared to controls and healthy C57 mice (J). (K–N) Quantification of tyrosinated (K,L) and detyrosinated microtubule (M,N) in liver sections further revealed differences across groups, highlighting the impact of combination therapy. In all graphs, results are shown as mean±s.d. a.u., arbitrary units. ***$P<0.001$; ns, not significant [unpaired two-tailed *t*-test (B,D,F,G,L,M); ordinary one-way ANOVA for Factor-FIX activity significance test (I)]. Scale bars: 20 µm (A,J,K,M); 100 µm (C).

or detyrosinated microtubules promote infection, AAV2 is impaired by detyrosinated microtubules, suggesting that non-enveloped viruses face distinct cytoskeletal barriers compared to enveloped viruses.

Previous studies show that nocodazole-induced microtubule depolymerization impairs AAV2 endosomal escape and infectivity (Xiao and Samulski, 2012). Nocodazole selectively targets labile microtubules at micromolar concentrations (Drabek et al., 2006; Guillaud et al., 1998), suggesting AAV2 retrograde transport is dependent on labile tyrosinated microtubules. Consistently, our findings show that increasing tyrosinated microtubule pools enhances AAV2 perinuclear trafficking and transduction, in contrast to HIV-1 and influenza A, which exploit stable microtubule populations (Delaney et al., 2017; Husain and Cheung, 2014; Husain and Harrod, 2011; Sabo et al., 2013). This divergence might stem from differences in viral entry receptors and the distinct mechanisms employed by enveloped and non-enveloped viruses to exploit the cytoskeleton to their advantage for endosomal sorting and trafficking to evade cellular degradation.

Mechanistically AAV2 enters cells via heparan sulfate proteoglycan (HSPG)-mediated endocytosis with growth factor receptors acting as co-receptors (Kashiwakura et al., 2005; Qing et al., 1999; Summerford and Samulski, 1998; Weller et al., 2010). Following internalization, HSPG is directed to lysosomal degradation, a process disrupted by vinblastine-induced microtubule depolymerization (Egeberg et al., 2001). Notably, vinblastine has been shown to depolymerize stable microtubules

marked by acetylation and impair autophagosome–lysosome fusion (Xie et al., 2010). Vinblastine is likely to affect all stable microtubules, including detyrosinated microtubules, suggesting that, upon AAV2 internalization, HSPG is directed to lysosomes along the detyrosinated microtubules. Furthermore, previous reports show that lysosomal motility is hindered on detyrosinated microtubules, enriching lysosomes on detyrosinated microtubules, facilitating autophagosome–lysosome fusion (Mohan et al., 2019). Taken together, these observations suggest that detyrosinated microtubules foster a lysosomal degradation hub where cellular cargoes and viral particles from endocytic and autophagic pathways accumulate proximal to lysosomes, facilitating their fusion and degradation. Congruently, we observe that detyrosinated microtubules accumulate AAV2, and their suppression disrupts AAV2 lysosomal entrapment.

Parthenolide and its pro-drug, DMAPT, are promising inhibitors of microtubule detyrosination (Fonrose et al., 2007; Gobrecht et al., 2024; Leibinger et al., 2023). Whereas the anticancer effects of parthenolide stem from NF-κB modulation (Hehner et al., 1999), its detyrosination inhibition at micromolar concentrations (Li et al., 2019) is independent of NF-κB activity (Fonrose et al., 2007). Parthenolide promotes axon regeneration in injury-induced mouse models (Gobrecht et al., 2024; Leibinger et al., 2023) and reduces muscle stiffness by modulating detyrosinated microtubule, offering therapeutic potential for muscular dystrophies (Kerr et al., 2015) and heart failure (Chen et al., 2018). In our hemophilia B mouse model, DMAPT enhanced AAV2-mediated human FIX gene delivery. These findings expand the potential applications of DMAPT beyond cardiac disease and cancer, highlighting its role in optimizing gene therapy outcomes. However, beyond suppressing microtubule detyrosination, parthenolide is also known to induce microtubule aggregation (Hotta et al., 2021) and exert antimitotic effects (Eibes et al., 2025). Thus, the effects of parthenolide observed in our study might potentially be due to detyrosination suppression or a combination of these actions, that need to be elucidated further.

In conclusion, we propose that microtubule PTMs regulate AAV2 trafficking and transgene expression (Fig. 6). AAV2 enters cells via receptor-mediated endocytosis (Duan et al., 1999; Nonnenmacher and Weber, 2011; Sanlioglu et al., 2000), and traffics through early and late endosomes (Ding et al., 2005) along microtubules mediated by dynein motor protein (Xiao and Samulski, 2012). Endosomal acidification activates the capsid phospholipase A2 domain, enabling membrane rupture and viral escape (Stahnke et al., 2011), leading to nuclear entry for gene expression. We demonstrate that AAV2 transport towards the nucleus is impaired on detyrosinated microtubules, causing lysosomal accumulation and limiting endosomal escape, resulting in reduced transgene expression. As AAV2 retrograde transport is mediated by dynein (Xiao and Samulski, 2012), the reduced motility observed on detyrosinated microtubules is consistent with previous reports showing that processivity of dynein–dynactin complexes are regulated by tyrosination status of tubulin (McKenney et al., 2016). Congruently, parthenolide treatment promotes rapid AAV2 transport along tyrosinated microtubules, enhancing endosomal escape near the nucleus, preventing cytoplasmic degradation, and boosting nuclear entry and transgene expression. Finally, these findings highlight the potential of a drug–vector co-administration strategy (co-administering parthenolide with AAV2) to enhance transgene expression and improve therapeutic outcomes for hemophilia B patients.

Our study is limited to the AAV2 serotype. Apart from validating the effects of detyrosination suppression across other AAV 1-10

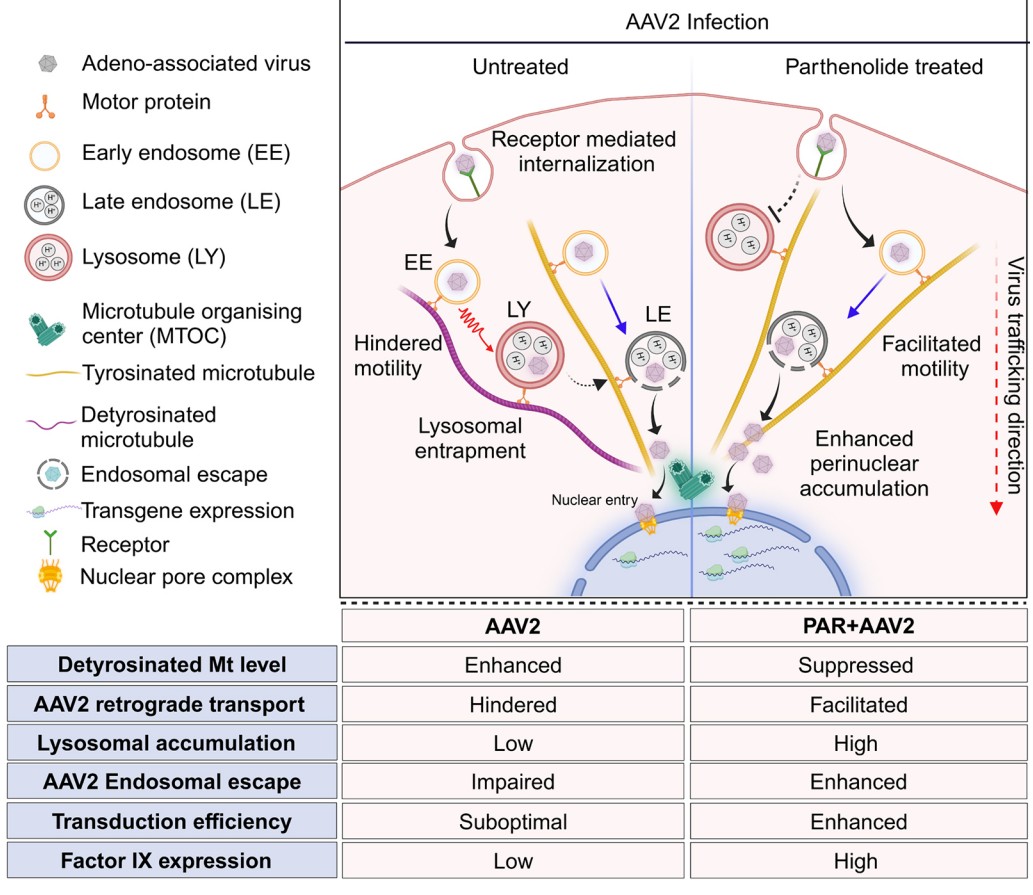

| | AAV2 | PAR+AAV2 |
|---|---|---|
| **Detyrosinated Mt level** | Enhanced | Suppressed |
| **AAV2 retrograde transport** | Hindered | Facilitated |
| **Lysosomal accumulation** | Low | High |
| **AAV2 Endosomal escape** | Impaired | Enhanced |
| **Transduction efficiency** | Suboptimal | Enhanced |
| **Factor IX expression** | Low | High |

**Fig. 6. Proposed model.** The figure illustrates AAV2 intracellular transport emphasizing the role of microtubule post-translational modifications in regulating AAV2 trafficking. It highlights the hindered motility of AAV2 on detyrosinated microtubules (red zigzag arrow), leading to its accumulation in lysosomes, with minimal endosomal escape. Conversely, parthenolide treatment (right panel) suppresses detyrosinated microtubule, enabling accelerated transport (blue arrows), increased endosomal escape close to the nucleus, leading to enhanced transduction efficiency. Created in BioRender by Jayandharan, G. R., 2025. https://BioRender.com/ayxuowi. This figure was sublicensed under CC-BY 4.0 terms.

serotypes, it will be useful to dissect the impact of non-canonical virus entry pathways on microtubule PTMs. Furthermore, a comprehensive evaluation of immune response to pharmacological modulation might be required. In this study, we focused on vasohibin as the detyrosination enzyme; however, AAV-induced detyrosination could also involve other enzymes, such as TMCP1 and TMCP2 (Nicot et al., 2023), which warrants further investigation. Interestingly, we also observed modulation of another microtubule PTM, polyglutamylation, with AAV2. Although beyond the scope of this study, polyglutamylated microtubules are known to regulate cargo transport; thus, clarifying their role in AAV-mediated gene delivery could yield valuable insights into how microtubule PTMs collectively govern viral trafficking and transduction. Nonetheless, our findings establish microtubule detyrosination as a key regulator of AAV2 trafficking and present a novel target for optimizing AAV-mediated gene therapy.

## MATERIALS AND METHODS
### AAV2 production and dye conjugation
AAV2 was prepared and produced as described previously (Gabriel et al., 2013; Mary et al., 2019a; Maurya et al., 2019). Briefly, recombinant AAV2 (scAAV2) vectors containing the human factor FIX (h.FIX) gene driven by the LP1 promoter/enhancer were generated by triple transfection protocol as described previously (Gabriel et al., 2013). Purified vectors were quantified by quantitative PCR using poly(A) region-specific forward and reverse

primers after DNase treatment (Aurnhammer et al., 2012). For the live imaging of AAV2, dye and virus were conjugated as described previously (Bartlett et al., 2000) with some modifications. Briefly AAV2 ($3\times10^{12}$ vgs; 20 µg) was incubated with Cy3 dye (Cytiva PA23001; 50 µM) in a conjugation buffer (0.1 M sodium bicarbonate, pH 9.3; Sigma-Aldrich) for 8 h at 4°C. The buffer was prepared by dissolving $NaHCO_3$ in water and adjusting the pH to 9.3 by gradually adding 0.1 M $Na_2CO_3$ while stirring. To remove unbound dye, the mixture was subjected to centrifugal filtration using 0.5 ml filters (Merck Millipore) in a dialysis buffer (10% glycerol, 10 mM Tris-HCl and 50 mM NaCl, pH 7.4 in deionized water). Centrifugation was performed at 14,000 $g$ for 15 min at 4°C, and the filtration process was repeated approximately eight times or until the solution appeared visibly clear, confirming the successful removal of unbound dye. The virus was then aliquoted and stored in −80°C. The degree of labeling of Cy3–AAV2 was determined by spectrophotometry, and it was speculated that the dye reacted with the ε-amine of lysine residues and the hydroxyl groups of serine and threonine. The labeled virus titer was determined by quantitative PCR.

### Cell culture and transfection
Human hepatoma-derived Huh7 (kind gift from Saumitra Das, Indian Institute of Science, Bengaluru, India) cells were maintained in Iscove's Modified Dulbecco's Medium (IMDM, Gibco) and African green monkey kidney epithelial cells, BSC1 (kind gift from Melike Lakadamyali, University of Pennsylvania, USA) were maintained in minimum essential medium (MEM, Gibco) supplemented with 2 mM L-glutamine and 1 mM sodium pyruvate. Media were supplemented with 10% fetal bovine serum

(FBS, Hyclone, USA), and 100 µg/ml penicillin-streptomycin (Invitrogen) or 100 µg/ml Anti-Anti (Gibco). All the cells were maintained at 37°C and 5% $CO_2$. For imaging experiments, cells were seeded in 4-well chambered, 8-well chambered or single-well imaging dishes (Ibidi, Gräfelfing, Germany) at ~$10^4$ cells per well with 200–400 µl medium. Plasmid transfection was performed using X-Treme GENE HP DNA transfection reagent (Sigma-Aldrich). Cells were transfected with 300 ng DNA for 24 h. pShuttle-mRFP-GSK3 S9A was Addgene #24371, pAd/CMV/V5-Clasp2α was Addgene #89763, pEGFP-CLASP2 512-650 9xS/A was Addgene #24410 (all deposited by Torsten Wittmann), Rab7a–GFP and Rab5a–GFP and LAMP1–GFP plasmids were a kind gift from Indranil Banerjee (IISER Mohali, India), VASH2–mCherry, VASH2 C158A–mCherry (dead mutant) and TTL–Scarlet, were a kind gift from Carsten Janke (Institut Curie, France), and VASH2–GFP and TTL-GFP were kindly provided by Minhaj Sirajuddin (InStem, Bengaluru, India), VASH1/2 shRNA and shRNA scramble were procured from IISc Bangalore shRNA consortium. See Table S1 for details of plasmids and shRNA.

### Pulse-chase treatment
For pulse-chase experiments, cells were incubated with adeno-associated virus serotype 2 (AAV2) at a multiplicity of infection (MOI) of $2×10^4$ ($2×10^4$ vgs/cell) in serum-free medium at 37°C for 2 h to facilitate viral adsorption onto the cell surface (Gerna et al., 1980; Hofmann and Wyler, 1988). Following incubation, unbound AAV2 was removed by washing the cells with phosphate-buffered saline (PBS). Cells were then incubated in medium supplemented with 10% FBS at 37°C for chase time points of 0, 2, 4 and 8 h. Mock-treated control cells underwent the same procedure without AAV2 exposure.

### Parthenolide treatment
Cells were incubated with various concentrations of parthenolide for 48 h, and cell viability was assessed using the MTT assay (see Fig. S4B) to determine the half-maximal inhibitory concentration ($IC_{50}$) of parthenolide (Sigma-Aldrich P0667) to be 100 µM. Additionally, Huh7 cells were treated with different concentrations of parthenolide to identify the optimal dose for suppressing detyrosinated microtubule, which was found to be 20 µM (see Fig. S4C). Further to that, in all experiments, cells were pretreated with 20 µM parthenolide for 2 h before the AAV2 treatment.

### Endosomal escape with Calcein
Cells were either untreated or pretreated with 20 µM parthenolide for 2 h. Subsequently, cells were treated with AAV2 for 2 h in serum-free medium as mentioned in pulse-chase treatment description above. Immediately after a medium change, Calcein (Sigma Aldrich C0875-5G) was added at a final concentration of 100 µg/ml in FBS-containing medium, marking the 0 h time point. Further cells were imaged at 0 h, 2 h, 4 h and 8 h post Calcein treatment using TIRF microscopy.

### Transduction assay
For transduction efficiency assays, cells were either left untreated or pretreated with 20 µM parthenolide for 2 h prior to infection. Cells were then infected with AAV2 encoding a GFP transgene at an MOI of $5×10^3$ vgs/cell or $2×10^4$ vgs/cell, in serum-free medium for 3 h. Following infection, the medium was replaced with FBS-containing medium, and cells were incubated for 48 h before fixation and imaging. Cells were permeabilized with 0.2% Triton X-100 for 2 min, stained with TO-PRO (Invitrogen T3605) for 10 min, and washed three times with 1× PBS. For flow cytometry analysis, cells were untreated or pretreated with 20 µM parthenolide for 2 h before infection and then infected with AAV2 at $5×10^3$ MOI for 48 h. GFP expression was quantified by flow cytometry (BD bioscience Accuri C6 plus) using 488 nm laser and 533/30 nm band pass filter.

### Immunofluorescence
Cells were fixed with either 4% PFA (15710, Electron Microscopy Science) or 3% PFA plus 0.2% glutaraldehyde (Millipore) for 10 min at room temperature (RT) and washed with 1× PBS (Gibco). A $NaBH_4$ solution (1 mg/ml) in PBS was used to quench the autofluorescence of GA. Cells were permeabilized and blocked with 3% BSA and 0.2% Triton X-100 in PBS for

2 h at RT. Cells were then incubated in primary antibodies for detyrosinated microtubule (ab48389; 1:100) or tyrosinated microtubule (ab6160; 1:100) or acetylated microtubule (Sigma Aldrich; T6793) or microtubule polyglutamylation (Adipogen; AB-20B-0020) for 2 h at RT (see Table S1 for full details of antibodies used). All the microtubule PTM antibodies chosen for our work were previously utilized in research works (Genova et al., 2023; Ho et al., 2023; Mitra et al., 2020; Mohan et al., 2019; Sabo et al., 2013). AAV2 (Fitzgerald 10R-A110a 1:100) or phospho-GSK-3β (Ser9) (CST 5558; 1:200) or CLASP2 (Invitrogen PA5-109547; 1:100) overnight at 4°C, and then washed three times with washing buffer (0.2% BSA and 0.05% Triton X-100) for 5 min each. Then incubated with dye-tagged secondary antibody, donkey anti-rabbit (Jackson ImmunoResearch, 711005152; 1:100), donkey anti-mouse-IgG (Jackson ImmunoResearch, 715005150; 1:100), donkey anti-rat-IgG (Jackson ImmunoResearch, 712005153; 1:100) in blocking buffer for 1.5 h. followed by three times wash with PBS for 5 min each. For STORM imaging secondary antibodies were labeled with Alexa Fluor 647 and Alexa Fluor 405 at ~1:4 ratio per antibody.

### Detyrosinated microtubule antibody titration
We used an antibody against detyrosinated microtubule that has been widely reported previously (Mitra et al., 2020; Mohan et al., 2019; Sabo et al., 2013). To validate its specificity in our experiments, we tested two cell lines: Huh7 cells, which show low baseline detyrosination, and BS-C-1 cells, which show high baseline detyrosination. Across antibody dilutions (1:25, 1:50, 1:100 and 1:1000) under control conditions (no AAV2 treatment), Huh7 cells showed negligible detyrosinated microtubules, while BS-C-1 cells displayed increasing signal intensity with antibody concentration, saturating at 1:100 (Fig. S1B), confirming the specificity and absence of cross-reactivity of the antibody.

### Immunoblotting
Samples lysis was done with Laemmli buffer (100 mM DTT, 2% SDS, 80 mM Tris-HCl pH 6.8, 10% glycerol and Bromophenol Blue). Samples were then denatured for 7 min at 95°C. The proteins were then separated by sodium dodecyl sulfate-polyacrylamide gel electrophoresis (SDS-PAGE) and transferred onto PVDF membranes. For immunoblotting, membrane blocking was done with 5% skimmed milk in TBS for 1 h at room temperature and then membranes were incubated with primary antibodies for α-tubulin (ab18251; 1:1000), tyrosinated microtubules (ab6160; 1:1000), detyrosinated microtubules (ab48389; 1:500), acetylated microtubules (Sigma-Aldrich; T6793; 1:1000), polyglutamylated microtubules (Adipogen; AB-20B-0020) and GAPDH (Invitrogen; 1:2000) and incubated overnight at 4°C. See Table S1 for details of antibodies. The next day membrane was washed with TBS plus 0.1% Tween 20 (TBST) three times for 5 min. The blots were then incubated with respective HRP-conjugated secondary antibodies prepared in TBST for 2 h at room temperature. After three washings with TBST for 5 min the membranes were incubated with Super signal west pico plus chemiluminescent substrate (Thermo Fisher Scientific, 34580; 1:1) to develop the bands. The band quantification was done using FIJI software (https://fiji.sc/). Fig. S6 shows all the original and uncropped images of the blots represented in Fig. 1E.

### Microscopy techniques
Confocal images were acquired using a Zeiss laser scanning confocal microscope (LSM 780 system) equipped with a 63× oil immersion, 40× oil immersion and 20× objectives. An argon laser 488 nm was used to excite GFP, a DPSS laser 561 nm was used to excite Cy3 and Alexa Fluor 560, and a HeNe laser 633 nm was used to excite Alexa Fluor 647.

The 3D super-resolution structured illumination microscopy (SIM) imaging was performed on fully motorized lattice SIM, ZEISS Elyra 7 (Carl Zeiss Ltd.) system equipped with Plan-Apochromat 63×/1.4 oil immersion objective lens, sCMOS camera and four filter sets with precisely mounted ACR-coded filter modules. A 488 nm laser was used to excite Alexa Fluor 488, a 561 nm laser was used to excite Alexa Fluor 560, and the 642 nm laser was used to excite Alexa Fluor 647 on a XY Piezo Scanning Stage. Image recoding was done on leap mode for the three times faster imaging, and the processing of the images was done by the in-built processing software of the Lattice SIM system.

TIRF microscopy was performed on a custom-built setup with a Nikon Ti2E microscope body, a 1.49 NA 100× oil immersion objective lens and an emission filter (ET705/72m; Chroma). Images were captured by an EM-CCD camera at an exposure time of 100 ms per frame for 1000 frames for live-cell imaging and 10 frames for fixed cell imaging.

A few images were acquired using SAFeRedSTORM module (Abbelight) mounted on an Evident/Olympus IX3 microscope equipped with an oil-immersion 100× objective, 1.5 NA (Evident/Olympus) and fiber-coupled 642 nm laser (450 mW Errol). Images were collected with an ORCA-Fusion sCMOS camera (Hamamatsu). Image acquisition and microscope control were driven by 586 NEO software (Abbelight).

### Correlative live-cell and STORM imaging

The live-cell imaging of AAV2–Cy3 was performed on the custom-built TIRF microscope described above. Images were acquired for ∼1000 frames at 100 ms exposure followed by *in situ* fixation of the cell. Cells were then immunostained for detyrosinated microtubule and the super resolution STORM image was acquired. A 647 nm laser (MPB communications) was used to excite the reporter dye (Alexa Fluor 647; Invitrogen) and a 405 nm laser (Obis; Coherent) was used to excite the 405-activator dye (Invitrogen). The emission of the light was collected by Nikon 100× oil immersion TIRF-SR objective (NA 1.49) and laser quad band set with emission filter (TRF89902-EMET-405/488/561/647 nm laser Quad Band Set for TIRF application; Chroma). Image localizations were captured by EM-CCD camera at an exposure time of 20 ms per frame for more than 60,000 frames for each STORM image. STORM images were analyzed and rendered as previously described, using custom-written software (Insight3, provided by Bo Huang, University of California, San Francisco, CA, USA; Huang et al., 2008).

To achieve channel registration, live-cell videos of AAV2 (Cy3 emission) excited with a 560-nm laser line and captured using a quad-band filter set were aligned with super-resolution images of detyrosinated microtubules (Alexa Fluor 647 emission) excited at 647 nm and acquired using the same filter set, following previously established methods (Mohan et al., 2019; Verdeny-Vilanova et al., 2017). Briefly, to account for sample drift during imaging, a rigid shift along the *x* and *y* axes was determined using TetraSpeck beads (Thermo Fisher Scientific, USA) placed on the imaging dish glass. These beads were visible in both the live-cell AAV2 images and the STORM images of detyrosinated microtubule. The calculated shifts were then applied to the raw localizations from the super-resolution imaging, using custom algorithms as described previously (Verdeny-Vilanova et al., 2017).

### Single-particle tracking

For single-particle tracking, AAV2 particle positions were tracked using custom-made, semi-automated particle tracking software. The *x* and *y* coordinates of the trajectories were extracted by fitting a 2D Gaussian function to the point spread function of the objects. To identify the active and passive transport phases, a MATLAB-based custom script was utilized, as described in previous studies (Verdeny-Vilanova et al., 2017). Briefly, the trajectory analysis involved a moving window approach, using four-point segments along the coordinate data points. For each segment, the ratio of the displacement between the segment starting and ending points to the sum of displacements between each point in the segment was computed. As segments overlapped, individual trajectory points contributed to multiple segments, and the ratio for each point was averaged across all its associated segments. This ratio served as a measure of linearity, with values near 1 indicating active transport. A threshold ratio of 0.7 was set to differentiate between active and passive phase. To refine the classification further, an angle-based criterion was applied – segments where consecutive displacement vectors formed angles smaller than 90° were labeled as passive. Additionally, mean square displacement (MSD) analysis was used to validate the classification, where active transport exhibited an α-value of greater than 1.5. Only trajectories with at least five data points were considered for further analysis. Metrics such as run length, speed, pause frequency per trajectory and pause duration were calculated from the processed data.

### Combination therapy in hemophilia B mice

For *in vivo* studies we utilized the pro-drug DMAPT (Abcam; ab146189) which is a fumarate salt of parthenolide with improved bioavailability. The animals were divided into mock, DMAPT only, scAAV2 LP1-h.FIX and xombination (DMAPT+scAAV2 LP1-h.FIX). Animals used in the study were 7–11 weeks old, with 5–7 animals per group. Animals in the DMAPT-only and combination groups were pre-treated with the DMAPT (20 μg/mouse) via oral gavage for 5 days before and 1 week after injection. The scAAV2 LP1-h.FIX and combination groups received a vector dose of $5×10^{10}$ vector genomes (vgs). Immediately before injections, the animals in the DMAPT-only and combination groups received another dose of the drug. Post-injection, DMAPT was administered on alternate days for up to 7 days. Blood samples were collected after 30 days using retro-orbital bleeding into tubes containing 3.8% sodium citrate (as an anticoagulant). Plasma was isolated using standard methods and stored at −80°C until analysis. FIX activity was measured using the Hyphen chromogenic FIX assay (Hyphen BioMed, Neuville-Sur-Oise, France), following the protocol provided by the manufacturer and as described earlier (Herrmann et al., 2020). Calibrators, control reagents, and buffers were brought to room temperature for 30 min before resuspension. They were then resuspended using Milli-Q water, mixed gently, and left undisturbed for 30 min. Standards were prepared by serial dilution, ranging from a high concentration of 200% to a low concentration of 3.125%. Sample dilutions were performed using the buffer provided with the kit. Readings were taken at 450 nm, and a standard curve was generated by plotting a straight line of log (OD) versus log (concentration). All animal experiments were performed according to approved guidelines [Animal Ethics Committee (IAEC). IAEC Protocol No.: IITK/IAEC/2022/1158].

### Immunohistochemistry

After 30 days of vector administration, animal liver tissue was harvested followed by blood collection (Zhou et al., 2022; Luhtala and Hunter, 2019). Liver tissue was fixed in 4% PFA in PBS overnight at 4°C. Sucrose gradient treatment was performed with 15% and 30% sucrose for 12 h and 2 h respectively. Tissue samples were further mounted in the PolyFreeze (Sigma-Aldrich) and cryo-sectioning was done (Leica CM1520, Leica Biosystems, Wetzlar, Germany) to obtain the 10 μm thick liver tissue sections. For immunohistochemistry, the tissue was then fixed again for 15 min in 4% PFA. The blocking was done using 5% normal donkey serum and 0.2% Triton X-100 in PBS for 2 h at RT. Primary antibody staining was performed against detyrosinated microtubule (1:100), tyrosinated microtubule (1:100) and Factor-IX protein (1:250) overnight at 4°C. Secondary antibody (1:500) incubation was done for 2 h at RT after PBS wash. This was followed by the nucleus staining with DAPI (Sigma-Aldrich) and slides were then mounted with Prolong Gold (Thermo Fisher Scientific, USA).

### Image processing and quantification

SIM images were processed using FIJI (https://fiji.sc/; Schindelin et al., 2012). Maximum intensity projection was applied to all images, and brightness and contrast were adjusted uniformly across all conditions to ensure consistent representation and quantification of microtubule-PTM ratio data. For intensity quantification, image thresholding was performed using identical parameters for all conditions, and the integrated density was measured. The ratio of detyrosinated and tyrosinated microtubule was then calculated by dividing their respective integrated densities. A similar approach has been previously established (Shen and Ori-McKenney, 2024). We performed the polyglutamylated and detyrosinated microtubule colocalization analysis using profile plot tool from ImageJ. To quantify the percentage of AAV2 associated with detyrosinated microtubules, multiple frames from each *z*-stack were analyzed. The percentage was calculated by dividing the number of AAV2 particles colocalized with detyrosinated microtubule by the total number of AAV2 particles. For track displacement analysis, we utilized the TrackMate (Ershov et al., 2022; Tinevez et al., 2017) plugin from FIJI software. For zonal distribution analysis, the outer boundary and nucleus of the cells were first outlined. Straight lines were then drawn from multiple points along the nuclear boundary to the outer boundary. Each line was divided into three equal segments, and corresponding boundaries were drawn to define three distinct regions – the outer zone (adjacent to the plasma membrane), the mid-zone (central cytoplasm) and the inner zone (proximal to the nucleus). The number of AAV2 particles within each zone was manually quantified. For

Rab7–AAV2 colocalization analysis, the colocalization of Rab7 with AAV2 was manually scored. For endosomal escape analysis using the Calcein assay, the integrated density was measured within the cytoplasmic area devoid of puncta, and values were compared across different conditions after subtracting the background integrated density.

## Statistical analysis

All the data was analyzed for significance using a two-sample unpaired two-tailed *t*-test and ordinary one-way ANOVA for *in vivo* studies. Where required data was normalized to a maximum value. $P \leq 0.05$ was considered statistically significant. All the analysis was performed using OriginPro 2021b (Academic).

## Acknowledgements

The vector production and FACS facility is partially supported by a grant (IA/TSG/22/1/600401) awarded by DBT-Wellcome Trust India Alliance G.R.J. We thank Prof. Melike Lakadamyali (University of Pennsylvania, Philadelphia, PA) for supporting the research with reagents and software. We thank Indranil Banerjee (IISER Mohali, India), Prof. Carsten Janke (Institut Curie, France) and Prof. Minhaj Sirajuddin (InStem, Bengaluru, India) for providing plasmids. We thank Prof. Appu Kumar Singh (IIT Kanpur) for critically reading the manuscript. N.M. thanks Prof. Pradip Sinha (IIT Kanpur) for suggestions and mentorship. The authors gratefully acknowledge the contribution of Jeganath A., Deepak M. Khushalani, and Sanket Patil in helping establish the laboratory facility at IIT Kanpur, India. S.T. acknowledges Dr. Wills Janardhanan (Zetlon Biotech Pvt. Ltd.) for guiding through western blot techniques. The authors also acknowledge Jemina J. for the help with single particle tracking. Furthermore, Pratiksha Sarangi has been noted for her help during animal studies. The authors acknowledge the Indian Institute of Technology, Delhi, SATHI facility and Mr Taseen Ahmad for Structured Illumination Microscopy imaging.

## Competing interests

The authors declare that IIT Kanpur has filed a patent application of the utilization of a co-therapy strategy and some technologies on AAV gene therapy has been licensed to commercial partners (G.R.J.).

## Author contributions

Conceptualization: G.R.J., N.M.; Data curation: S.T., D.C., G.R.J., N.M.; Formal analysis: S.T., G.R.J., N.M.; Funding acquisition: G.R.J., N.M.; Investigation: S.H., G.R.J., N.M.; Methodology: S.T., S.H., J.K., D.C., G.R.J., N.M.; Project administration: G.R.J., N.M.; Resources: G.R.J., N.M.; Software: G.R.J., N.M.; Supervision: G.R.J., N.M.; Validation: S.T., S.H., D.C., G.R.J., N.M.; Visualization: G.R.J., N.M.; Writing – original draft: S.T., G.R.J., N.M.; Writing – review & editing: G.R.J., N.M.

## Funding

S.T. expresses gratitude towards the Council of Scientific and Industrial Research, India for providing financial assistance during research. N.M. and G.R.J. acknowledge funding from the Indian Council of Medical Research, India (ICMR F.No. 2021-15239/GTGE/Adhoc-BMS) and the Indian Institute of Technology Kanpur, India for the present study. G.R.J. is supported by a Team Science grant awarded by the DBT-Wellcome Trust India Alliance (IA/TSG/22/1/600401). Open Access funding provided by Indian Institute of Technology Kanpur. Deposited in PMC for immediate release.

## Data and resource availability

The data set studied and investigated during the current study are accessible from the corresponding authors (N.M. and G.R.J.) upon reasonable request. All relevant data and details of resources can be found within the article and its supplementary information.

## First Person

This article has an associated First Person interview with the first author of the paper.

## Peer review history

The peer review history is available online at https://journals.biologists.com/jcs/lookup/doi/10.1242/jcs.264190.reviewer-comments.pdf

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
