## [Peer Review File · Journal of Cell Science]

Suppressing microtubule detyrosination augments AAV2 endosomal escape and gene delivery

Shefali Tripathi, Shamshul Huda, Joydipta Kar, Dinesh Chandra, Giridhara R. Jayandharan and Nitin Mohan

DOI: 10.1242/jcs.264190

Editor: Michael Way

Review timeline

Original submission:	4 June 2025
Editorial decision:	9 July 2025
First revision received:	22 September 2025
Editorial decision:	9 October 2025
Second revision received:	16 October 2025
Accepted:	17 October 2025

Original submission

First decision letter

MS ID#: jcs.264190

MS TITLE: Suppressing microtubule detyrosination augments AAV2 endosomal escape and gene delivery.

AUTHORS: Shefali Tripathi; Shamshul Huda; Joydipta Kar; Dinesh Chandra; Jayandharan R Giridhara; Nitin Mohan

ARTICLE TYPE: Research Article

Dear Dr Mohan,

We have now reached a decision on the above manuscript.

To see the reviewers' reports and a copy of this decision letter, please go to:

As you will see, both reviewers are positive about your study. However, they raise a number of points that prevent me from accepting the paper at this stage. They suggest, however, that a revised version might prove acceptable, if you can address their concerns with additional experiments. If you think that you can deal satisfactorily with the criticisms on revision, I would be pleased to see a revised manuscript. We would then return it to the reviewers.

Reviewer 1

Advance summary and potential significance to field

Tripathi et al. explore a novel role for microtubule detyrosination in the infection of cells with AAV. They present data indicating that microtubules become detyrosinated in a GSK3 β -dependent manner upon viral infection. A consequence of this appears to be a reduction in the motility of endocytosed virions. Subsequent manipulation of tubulin tyrosination levels using both

overexpression of tubulin-modifying enzymes and small-molecule inhibition of detyrosination supports these conclusions. This is linked to a role in endosomal escape, as shown through imaging of markers of the endolysosomal system, where detyrosination promotes lysosomal accumulation and presumably degradation. Additional experiments *in vitro* and in a model of AAV transduction suggest that inhibition of detyrosination enhances efficacy, leading the authors to conclude that modulation of this host response to infection might offer a means to improve AAV-based therapies. Overall, I found the manuscript to be novel and interesting, and the system/model is probed with a variety of tools and approaches that broadly converge on similar conclusions. The correlative live-cell/STORM imaging is particularly nice. The manuscript is quite well written, but in line with some comments below, the authors could more clearly explain the rationale for specific experiments and cite appropriate literature throughout, and the manuscript would benefit considerably from orthogonal validation of the initial findings.

Major Points:

1. Given the centrality of the observation to the authors' findings, I would have expected to see the initial results reported in Figure 1—AAV-induced detyrosination—confirmed using an orthogonal approach. A time-course western blot for detyrosinated tubulin would seem feasible, given the approximately fourfold change expected from imaging assays in Huh7 cells. I would generally suggest that where possible, the effects of manipulation of tyrosination are also confirmed by western blot.
2. The validation of the tubulin antibodies used in the system here is absolutely critical. This does appear, partly, in Figure S3, but gets lost, I would suggest that elements of this are brought into Figure 1, so that the reader has confidence in the author's approach from the outset. The authors might also cite additional reference highlighting where these (2x Abcam) reagents have been used in similar contexts and the reason for their choice as the product datasheets don't give much insight.

Minor Points:

1. Figure 1B, detyrosinated tubulin, 8 hr panel: This signal strongly resembles nuclear envelope staining. If this is representative, it should be commented on; if not, a more representative image might be preferable.
2. CLASP2 experiments: The narrative does not clearly convey the relevance or interpretation of the CLASP2 data. While I understand that CLASP2 has been shown to bind preferentially to detyrosinated microtubules, is there evidence that CLASP2 directly regulates tyrosination, as implied in line 38: "GSK3¹ as a key negative regulator of CLASP2-mediated detyrosination"?
3. The authors should cite relevant literature when introducing the various manipulations (e.g., VASH overexpression, TTL overexpression, parthenolide treatment). For example, Fonrose et al. 2007 could be cited to support the use of parthenolide. In addition, the limitations of parthenolide should be acknowledged—it likely acts indirectly by promoting tubulin aggregation (see 10.1016/j.cub.2020.11.055).
4. Building on the point above, I believe it would be premature to conclude that the effects observed in the mouse model are due directly to detyrosination. The drug likely has many effects on the organism. The authors should consider tempering their conclusions accordingly. The experiment is interesting, but I don't think this strong translational claim is critical for a mechanistic cell biology journal.
5. Figure 1 quantification: If I understand the approach correctly, cells are co-stained for tyrosinated and detyrosinated tubulin, signals are thresholded and quantified by integrated density, summed, and percentages calculated. This seems broadly reasonable, but I would caution that this does not necessarily provide an absolute value for percentage tyrosination/detyrosination, as implied. The relative signal would depend on the antibodies' respective efficacies, which have not been calibrated. I would suggest framing this as a "ratio of tyrosinated to detyrosinated signal" rather than an absolute percentage.

6. I may have missed them, but I could not find any supplementary movie legends.
7. The authors' model appears broadly consistent with an effect of detyrosination on reducing dynein activity? If so, this would align with the findings of McKenney and colleagues (<https://pubmed.ncbi.nlm.nih.gov/26968983/>). Could the potential mechanism be expanded upon in the discussion?

Reviewer 2

Advance summary and potential significance to field

The manuscript of Tripathi et al describes the role of the posttranslational tubulin modification detyrosination in the modulation of the efficiency of virus transduction with a focus on AAVs. AAVs are promising vectors for gene therapy. One of the bottlenecks in their efficiency is their intracellular translocation to the nucleus and their endosomal escape. Because posttranslational modifications of tubulin (i.e. microtubules) have recently been shown to control various intracellular processes including transport, the authors investigated their roles in virus infection. Based on superresolution microscopy methods that are executed with high quality, the authors show that microtubule detyrosination increases upon AAV2 infection of cells, which in turn reduces the efficiency of virus transduction. To show that modulation of detyrosination levels could help to overcome this, they lowered tubulin detyrosination levels in infected cells by different experimental means. The main conclusion of the manuscript is that increase of microtubule detyrosination in cells infected by AAV2 virus is lowering infection efficiency, and consequently increase of tyrosination (i.e. decrease or inhibition of detyrosination) could be used as a mean to improve virus infection.

Apart from this key finding, the authors dissect signalling pathways involved in the alteration of detyrosination in infected cells and perform a proof-of-principle study in mice.

Overall, this is a high-quality study providing solid evidence for most the claims and conclusions made. Yet, a number of points need to be addressed before the manuscript can be considered for publication.

Major points:

- 1) The entire paper is based on the assumption that increase of detyrosination is responsible for the observed effects on vesicle transport and reduced virus infection efficiency. Given the importance of this claim, the authors must provide complementary evidence for the specificity of this effect to detyrosination. This includes a western blot of control vs infected cells with a panel of tubulin modification-specific antibodies to show the degree of changes in detyrosination, tyrosination and other tubulin modification. This is particularly important as recent work has shown a key role in tubulin glutamylation in transport control, and as the authors mention themselves, changes in acetylation have been observed following virus infection. They mention that acetylation is undetectable, but perhaps there is a weak signal that can be detected and changes during virus infection? Most importantly, (poly)glutamylation must be carefully assessed given its previously shown impact on transport.
- 2) Part of the functional assays to demonstrate the causative role of detyrosination in the observed phenomena are based on the overexpression of tubulin tyrosine ligase (TTL). TTL has been shown to increase tyrosination in cells with dynamic microtubules - because it can only modify tubulin dimers, but not the polymerised microtubules. It is thus important to make sure that the detyrosinated microtubules the authors identified in AAV-infected cells are still dynamic, given that only tubulin from dynamic microtubules can be re-tyrosinated.
- 3) Another functional assay is the inhibition of detyrosinase activity with parthenolide - a drug that was initially thought to inhibit tubulin detyrosinating enzymes, which however was recently been shown to not be the case (Hotta et al. (2021) Parthenolide Destabilizes Microtubules by Covalently Modifying Tubulin. *Curr Biol* 31: 900-907 e906; Eibes et al. (2025) Parthenolide disrupts mitosis by inhibiting ZNF207/BUGZ-promoted kinetochore-microtubule attachment. *EMBO J* 44: 3764 - 3793). Thus, while past studies have shown that treatment with parthenolide can alter tubulin detyrosination, the new knowledge on the mechanism of action of this drug casts some

doubts on the straightforward link between its effects and this tubulin modification. The authors should thus consider using alternative experimental approaches, such as RNA interference for Vash proteins (detyrosinating enzymes), to more directly interfere with tubulin detyrosination levels in cells.

4) In Fig 2, the authors quantify the association of AAV2 particles with detyr- and tyr-microtubules, but they only show detyr-microtubules (similar in the suppl. Movies). If they have quantified association with tyr-microtubules, as Fig 2H,I,J,K suggest, then they must show representative images (and movies) for these experiments.

5) Movies 1, 2: why do some particles cross the detyr-microtubules without stopping there? Some videos stop abruptly, without showing what happens with the particle next (but concluding that it stays on the detyr-microtubules). The authors must provide an explanation for this, and should show more complete videos.

Minor points:

1) For better readability, all plots and graphs in the figures should carry a short descriptive title above the panels.

2) It is not clear in Fig 1D,E how the fraction of detyr- and tyr-microtubules was determined - did the authors classify microtubules into these two categories, and then measure their overall length, or did they simply quantify the fluorescent signal of the staining. As mentioned in the main points, an accompanying western blot would be very helpful here.

3) Fig 1J,L,N - it is difficult to see the detyr-tubulin staining in the merged images - the authors should show monochrome panels, at least for the detyr-staining, to allow the reader to get a clear picture of the differences between the represented cells.

4) The authors assume that the detyrosinating enzyme responsible for the observed change in tubulin modification is Vash - but did they verify this? There are two Vash enzymes, and two additional enzymes from a different family (MATCAP also known as TMCP1, and the homologue TMCP2; Nicot et al. (2023) A family of carboxypeptidases catalyzing alpha- and beta-tubulin tail processing and deglutamylation. *Sci Adv* 9: eadi7838). Can the authors provide evidence that the Vash enzyme they propose to be in charge of detyrosination after AAV2 infection is indeed Vash1?

5) Have the experiments based on overexpression of tubulin-modifying enzymes been controlled by overexpression of enzymatically dead enzymes? The paper shows no evidence of this. However, these controls are important as they exclude overexpression artefacts.

6) Correct bugs in figures: 4D, 4G, 5D, 5E, 5L.

7) Add cell line name in figure panels if different ones represented in a figure (ex Fig 1D, E).

First revision

Author response to reviewers' comments

Dear Editor,

Please find below our detailed, point-by-point responses to each of the reviewers' comments.

Reviewer 1

Major Points:

1. Given the centrality of the observation to the authors' findings, I would have expected to see the initial results reported in Figure 1—AAV-induced detyrosination—confirmed using an orthogonal approach. A time-course western blot for detyrosinated tubulin would seem feasible, given the approximately fourfold change expected from imaging assays in Huh7 cells. I would generally suggest that where possible, the effects of manipulation of tyrosination are also confirmed by western blot.

We thank the reviewer for emphasizing the importance of orthogonal approaches to confirm AAV-induced changes in microtubule detyrosination. To address this, we performed western blot analysis on mock- and AAV2-treated cells (4 hours post-infection), assessing multiple microtubule PTMs, including detyrosination, tyrosination, acetylation, and polyglutamylation. The 4-hour time point was chosen because our imaging data showed maximal detyrosination at this stage.

For quantification, blot intensities of each PTM were normalized to GAPDH, and fold changes were calculated relative to mock controls. Statistical significance was determined from three independent biological replicates. Consistent with our SIM imaging results, AAV2 treatment caused a ~4-fold increase in detyrosinated tubulin and a modest decline in tyrosination. Acetylation levels remained unchanged, whereas polyglutamylation increased 1.7-fold (Figures 1E, 1F).

These new results have been incorporated into the main text (lines 145-149), included in the main figures (Figures 1E and 1F), and supported by supplementary data (Figure S6). The methods section has also been updated to describe the western blot protocol and quantification procedures (line number 482-495).

2. The validation of the tubulin antibodies used in the system here is absolutely critical. This does appear, partly, in Figure S3, but gets lost, I would suggest that elements of this are brought into Figure 1, so that the reader has confidence in the author's approach from the outset. The authors might also cite additional references highlighting where these (2x Abcam) reagents have been used in similar contexts and the reason for their choice as the product datasheets don't give much insight.

We thank the reviewer for raising this important point. To thoroughly validate the antibody against detyrosinated microtubules, we tested it at multiple dilutions (1:25, 1:50, 1:100, and 1:1000) in two cell lines, Huh7 and BS-C-1, under basal (non-infected) conditions. As expected, Huh7 cells, which exhibit very low levels of polymerized detyrosinated microtubules, did not show detectable staining at any dilution. In contrast, BS-C-1 cells, known to have higher basal detyrosination, displayed robust staining of polymerized detyrosinated microtubules, with saturation at the 1:100 dilution, thereby confirming the detection of this microtubule subset.

To further confirm specificity, we performed shRNA-mediated knockdown of vasohibins (VASH2), the enzymes responsible for detyrosination, in both Huh7 and BS-C-1 cells. Immunostaining with the same antibody showed a complete loss of detyrosinated microtubule signal in shRNA-treated cells, while control cells retained strong staining (Figure 5A, Figure S1C, Figures S5A-D).

Together, these results demonstrate the specificity of the antibody for detyrosinated microtubules. Furthermore, this antibody has been widely used in previous studies to specifically detect detyrosinated tubulin, and we have now cited the relevant publications (Genova

et al., 2023; Ho et al., 2023; Mitra et al., 2020; Mohan et al., 2019; Sabo et al., 2013). These results are included in Figure S1B.

These new results have been incorporated into the main text (line number 140-142), and supported by supplementary data (Figure S1B, S1C and). The methods section has also been updated to describe the antibody titration and validation procedures (line number 474-781).

Minor Points:

1. Figure 1B, detyrosinated tubulin, 8 hr panel: This signal strongly resembles nuclear envelope staining. If this is representative, it should be commented on; if not, a more representative image might be preferable.

This observation is consistent throughout the 8-hour infected cells. This could be because most of the detyrosinated microtubule filaments in the cytoplasm have disappeared, with only a small fraction near the nucleus and those wrapping around the nucleus remaining visible, giving the impression of a nuclear envelope stain. We have included a statement in the figure legend to comment on this observation (line number 889-890)

2. CLASP2 experiments: The narrative does not clearly convey the relevance or interpretation of the CLASP2 data. While I understand that CLASP2 has been shown to bind preferentially to detyrosinated microtubules, is there evidence that CLASP2 directly regulates tyrosination, as implied in line 38: "GSK3B as a key negative regulator of CLASP2-mediated detyrosination"?

We agree with the reviewer that there are no prior evidence suggesting that CLASP2 can directly regulate tyrosination. Hence we have toned down our interpretations and explained our results with better clarity in the main text (line numbers 178-187), as the following.

“Our data indicates that merely overexpressing wild-type CLASP2 does not cause detyrosination. In contrast, overexpression of a Gsk3B-insensitive mutant of CLASP2 does induce detyrosination. This suggests that Gsk3B might inhibit CLASP2 binding to microtubules. When treated with AAV2, RTK signaling is activated, leading to the inhibition of Gsk3B and allowing CLASP2 to associate with microtubules. This binding of CLASP2 could play a role in promoting microtubule detyrosination through mechanisms that are still to be investigated. This mechanism also supports the host pathogen interaction driven pathway as reported previously that HIV induces microtubule stabilization by exploiting the CLASP2 (Mitra et al., 2020). Further, GSK3B may also regulate VASH-induced detyrosination through CLASP2-independent mechanisms, which warrants further investigation.”

3. The authors should cite relevant literature when introducing the various manipulations (e.g., VASH overexpression, TTL overexpression, parthenolide treatment). For example, Fonrose et al. 2007 could be cited to support the use of parthenolide. In addition, the limitations of parthenolide should be acknowledged—it likely acts indirectly by promoting tubulin aggregation (see 10.1016/j.cub.2020.11.055).

We thank the reviewer for pointing this out. We have now added relevant references supporting the use of VASH overexpression (Aillaud et al., 2017; Ramirez-Rios et al., 2023; Tang et al., 2023), TTL overexpression (Pietsch et al., 2024; Tang et al., 2023), and parthenolide treatment (Fonrose et al., 2007), including Fonrose et al. (2007) for parthenolide. We also acknowledge the limitations of parthenolide, which has been reported to act indirectly by promoting tubulin aggregation (Hotta et al., 2021) and have highlighted this caveat in the discussion section of the main text (line number 359-362).

“However, beyond suppressing microtubule detyrosination, parthenolide is also known to induce microtubule aggregation (Hotta et al., 2021) and exert antimitotic effects (Eibes et al., 2025). Thus, the effects of parthenolide observed in our study may potentially be due to detyrosination suppression or a combination of these actions, that need to be elucidated further.”

4. Building on the point above, I believe it would be premature to conclude that the effects observed in the mouse model are due directly to detyrosination. The drug likely has many effects on the organism. The authors should consider tempering their conclusions accordingly. The experiment is interesting, but I don't think this strong translational claim is critical for a mechanistic cell biology journal.

We fully agree with the reviewer that it is premature to attribute the enhanced FIX activity observed in the mouse model solely to detyrosination. We have toned down conclusion of the results as the following in the main text (line number 298-304).

“However, the enhanced FIX activity observed in the mouse model cannot solely be attributed to detyrosination suppression given the pleiotropic effects of parthenolide. Additional studies are required to validate the drug and vector dosage, as well as observed phenotypic effect during suppression of microtubule detyrosination. Nonetheless, our in vitro and in vivo results together demonstrate that suppressing microtubule detyrosination—a host-cell barrier that traps AAV2 on retrograde transport pathways—enhances viral nuclear entry and gene delivery efficacy, offering a clinically actionable strategy to optimize AAV-based therapies.”

Further, we have acknowledged these limitations of parthenolide and outlined this as an important direction for future investigation in the discussion section (line number 359-362), as the following.

“However, beyond suppressing microtubule detyrosination, parthenolide is also known to induce microtubule aggregation (Hotta et al., 2021) and exert antimitotic effects (Eibes et al., 2025). Thus, the effects of parthenolide observed in our study may potentially be due to detyrosination suppression or a combination of these actions, that need to be elucidated further.”

5. Figure 1 quantification: If I understand the approach correctly, cells are co-stained for tyrosinated and detyrosinated tubulin, signals are thresholded and quantified by integrated density, summed, and percentages calculated. This seems broadly reasonable, but I would caution that this does not necessarily provide an absolute value for percentage tyrosination/detyrosination, as implied. The relative signal would depend on the antibodies' respective efficacies, which have not been calibrated. I would suggest framing this as a "ratio of tyrosinated to detyrosinated signal" rather than an absolute percentage.

We thank the reviewer for this important clarification. We agree that absolute percentage quantification is not possible due to differences in antibody efficacy. To address this, we have reframed our analysis as a ratio of detyrosinated to tyrosinated microtubule signal. Specifically, we applied intensity thresholding, measured the integrated density of detyrosinated microtubules, and divided this by the corresponding tyrosinated microtubule signal to obtain the ratio. We have corrected the Y-axis labeling in Figure 1D and Figure S11 to reflect this change, added a schematic illustrating our quantification strategy in the supplementary results (Figure S1A), and cited relevant literature supporting this approach (Shen and Ori-McKenney, 2024).

6. I may have missed them, but I could not find any supplementary movie legends.

We apologize for not attaching the supplementary video legend earlier. We have now included it in the revised version in the supplementary document.

Supporting videos

Video 1. The video demonstrates the correlative imaging of AAV2 (green) in live-cell conditions and STORM imaging of detyrosinated tubulin (magenta). The real-time dynamics of AAV2 were observed one hour post-infection, with its trajectory displayed as a yellow track over the video, highlighting the movement of viral particles as they interact with the microtubule. AAV2 motility is notably restricted when encountering detyrosinated tubulin.

Video 2. The majority of virus particles remain stalled on detyrosinated tubulin throughout the duration of the movie, highlighting its restrictive effect.

Video 3. In contrast, when the virus moves along microtubule pathways that do not contain detyrosinated tubulin, its motility is unimpeded, allowing for faster movement.

Videos 4 and 5. AAV2 trafficking on tyrosinated microtubule reveal that virus moves along the tyrosinated microtubule unimpeded.

Video 6. AAV2 Tracks derived from Trackmate analysis of Huh7 cells in control conditions.

Video 7. AAV2 Tracks in Huh7 cells overexpressing VASH2.

Video 8. AAV2 Tracks in Huh7 cells overexpressing TTL

Video 9. AAV2 Tracks in Huh7 cells pretreated with parthenolide.

7. The authors' model appears broadly consistent with an effect of detyrosination on reducing dynein activity? If so, this would align with the findings of McKenney and colleagues (<https://pubmed.ncbi.nlm.nih.gov/26968983/>). Could the potential mechanism be expanded upon in the discussion?

Yes, it is revealed by McKenny group that the tyrosination status of the microtubules promotes dynein processivity. As suggested, we have discussed the possibility of dynein mediated transport of AAV being hindered by detyrosination in the main text (line number 370-373) and cited the relevant literature (McKenney et al., 2016).

“Since AAV2 retrograde transport is mediated by dynein, the reduced motility observed on detyrosinated microtubules is consistent with previous reports showing that processivity of dynein-dynactin complexes are regulated by tyrosination status of tubulin.”

Reviewer 2

Major Points:

The entire paper is based on the assumption that increase of detyrosination is responsible for the observed effects on vesicle transport and reduced virus infection efficiency. Given the importance of this claim, the authors must provide complementary evidence for the specificity of this effect to

detyrosination. This includes a western blot of control vs infected cells with a panel of tubulin modification-specific antibodies to show the degree of changes in detyrosination, tyrosination and other tubulin modifications. This is particularly important as recent work has shown a key role in tubulin glutamylation in transport control, and as the authors mention themselves, changes in acetylation have been observed following virus infection. They mention that acetylation is undetectable, but perhaps there is a weak signal that can be detected and changes during virus infection? Most importantly, (poly)glutamylation must be carefully assessed given its previously shown impact on transport.

We thank the reviewer for emphasizing the importance of complementary approaches to confirm AAV-induced changes in microtubule PTMs. To address this, we performed western blot analysis on mock- and AAV2-treated cells (4 hours post-infection), assessing multiple microtubule PTMs, including detyrosination, tyrosination, acetylation, and polyglutamylation. The 4-hour time point was chosen because our imaging data showed maximal detyrosination at this stage.

For quantification, blot intensities of each PTM were normalized to GAPDH, and fold changes were calculated relative to mock controls. Statistical significance was determined from three independent biological replicates. Consistent with our SIM imaging results, AAV2 treatment caused a ~4-fold increase in detyrosinated tubulin and a modest decline in tyrosination. Acetylation levels remained unchanged, whereas polyglutamylation increased 1.7-fold (Figures 1E, 1F).

The basal level of microtubule acetylation in Huh7 cells is minimal, as reported previously (Li et al., 2020; Lu et al., 2007). Given that AAV2 exhibits strong tropism for liver cells, we selected Huh7 cells as the primary model for this study. It is possible that other AAV serotypes or different viruses in alternative cell types may influence acetylation, suggesting that our findings on acetylated microtubules are likely cell type specific.

We examined polyglutamylation in Huh7 and BS-C-1 cells using confocal microscopy (Figure X). In Huh7 cells, we detected only a modest increase in PolyE signal, but clear visualization of distinct PolyE microtubules was challenging. By contrast, BS-C-1 cells displayed considerable PolyE microtubules under basal conditions, with the majority colocalizing with detyrosinated microtubules (Figure Y). Because AAV2 treatment induced far stronger changes in detyrosination compared to polyglutamylation, and given their substantial colocalization, it is difficult to disentangle the individual contribution of PolyE from that of detyrosination.

While we recognize the importance of elucidating the role of polyglutamylation in AAV2 trafficking and gene delivery, such investigations require extensive experimental approaches to selectively manipulate this modification, which lies beyond the scope of the current work. We have highlighted this point in the discussion as a future research direction (line number 381-387) and in the result section (line number 149-152).

The new results have been incorporated into the main text (lines 145-154), included in the main figures (Figures 1E and 1F), and supported by supplementary data (Blot transparency Figure S6). The methods section has also been updated to describe the western blot protocol and quantification procedures (line number 482-495).

2) Part of the functional assays to demonstrate the causative role of detyrosination in the observed phenomena are based on the overexpression of tubulin tyrosine ligase (TTL). TTL has been shown to increase tyrosination in cells with dynamic microtubules, because it can only modify tubulin dimers, but not the polymerized microtubules. It is thus important to make sure that the detyrosinated microtubules the authors identified in AAV2-infected cells are still dynamic, given that only tubulin from dynamic microtubules can be re-tyrosinated.

We thank the reviewer for raising this important point. To directly test whether AAV2-induced detyrosinated microtubules are dynamic and therefore subject to re-tyrosination by TTL, we examined their persistence under TTL overexpression. If these microtubules were highly dynamic, TTL should re-tyrosinate the released tubulin dimers, leading to a loss of detyrosinated microtubules. However, we consistently observed the presence of detyrosinated microtubules even after TTL overexpression, indicating that they are not efficiently re-tyrosinated and are therefore unlikely to be highly dynamic. As expected, TTL overexpression significantly increased tyrosinated microtubules and overall polymerized α -tubulin levels. This suggests that while the pre-existing detyrosinated microtubules remain stable, an expanded pool of tyrosinated microtubules provides more permissive tracks for AAV2, thereby enhancing its motility and transduction efficiency.

Supporting this interpretation, our live-cell and STORM imaging show that AAV2 tends to pause on detyrosinated microtubules but resumes movement upon switching to tyrosinated ones. Thus, with TTL overexpression, the greater abundance of tyrosinated tracks likely minimizes the barrier posed by detyrosinated microtubules. Furthermore, our data with CLASP2 reveal that it colocalizes with AAV2-induced detyrosinated microtubules. Given CLASP2's established role in stabilizing microtubules (Drabek et al., 2006), this further suggests that these detyrosinated microtubules are relatively stable rather than highly dynamic.

We have included this new perspective in the revised text (line number 227-231) and provided supporting results showing persistence of detyrosinated microtubules upon TTL overexpression in the supplementary data (Figure S3D-F).

3) Another functional assay is the inhibition of detyrosinase activity with parthenolide - a drug that was initially thought to inhibit tubulin detyrosinating enzymes, which however was recently been shown to not be the case (Hotta et al. (2021) Parthenolide Destabilizes Microtubules by Covalently Modifying Tubulin. *Curr Biol* 31: 900-907 e906; Eibes et al. (2025) Parthenolide disrupts mitosis by inhibiting ZNF207/BUGZ-promoted kinetochore- microtubule attachment. *EMBO J* 44: 3764 - 3793). Thus, while past studies have shown that treatment with parthenolide can alter tubulin detyrosination, the new knowledge on the mechanism of action of this drug casts some doubts on the straightforward link between its effects and this tubulin modification. The authors should thus consider using alternative experimental approaches, such as RNA interference for Vash proteins (detyrosinating enzymes), to more directly interfere with tubulin detyrosination levels in cells.

We thank the reviewer for this insightful suggestion. To directly address the concern regarding parthenolide, we employed an orthogonal approach by knocking down VASH1 or VASH2—the

enzymes responsible for detyrosination—using shRNAs. In Huh7 cells, VASH knockdown markedly reduced detyrosinated microtubules and significantly enhanced AAV2 transduction, as reflected by strong EGFP expression. Quantification showed ~80% of VASH2 shRNA-treated cells were successfully transduced (Figure 5A), while VASH1 knockdown also resulted in elevated AAV2-GFP expression (Figure S5A and S5B), suggesting that both enzymes contribute to the AAV2-induced increase in detyrosination.

Because Huh7 cells have very low basal detyrosination, we further validated these findings in BS-C-1 cells, which display ~23% detyrosinated tubulin at baseline. Consistently, VASH2 knockdown in BS-C-1 cells also enhanced AAV2 transduction (Figure S5C and S5D). Together, these results provide direct genetic evidence that suppression of VASH enzymes reduces detyrosination and enhances AAV2 transduction, thereby confirming the causal role of detyrosination beyond the pleiotropic effects of parthenolide.

These results have been incorporated into the revised text (line number 271-278), Figure 5A, Figure S5A-D, and methods (line number 424-425).

4) In Fig 2, the authors quantify the association of AAV2 particles with detyr- and tyr-microtubules, but they only show detyr-microtubules (similar in the suppl. Movies). If they have quantified association with tyr- microtubules, as Fig 2H, I, J, K suggest, then they must show representative images (and movies) for these experiments.

We thank the reviewer for bringing this to our attention. Previously we assigned AAV that were not moving on detyrosinated microtubules as the ones associated to tyrosinated microtubules. We have now performed correlative experiments with tyrosinated microtubules and deduced the motility parameters of AAV moving on tyrosinated microtubules. The representative images (Figure. S2G and S2J) and movies (supplementary video 4 and 5) have been included. The motility parameters have been modified in Figure 2H-K.

5) Movies 1, 2: why do some particles cross the detyr-microtubules without stopping there? Some videos stop abruptly, without showing what happens with the particle next (but concluding that it stays on the detyr-microtubules). The authors must explain this, and should show more complete videos.

Movies 1 and 2 show that the AAVs already moving on non-detyrosinated microtubules exhibited pausing behavior when they encountered detyrosinated microtubules, with extended pause durations. For Movie 1 we do not have the extended time period video that showed what happens to AAV after encountering detyrosinated microtubule, hence we removed it from our representations. However, Movie-2 showed that after the AAV pause at the detyrosinated microtubules, some of the AAVs resumed their journey by switching back to non-detyrosinated (presumably tyrosinated), microtubules. Hence, we replaced Movie-1 with Movie-2 to give a more continuous representation. The representative movie for AAVs that stayed continuously on a detyrosinated microtubule are shown in video 2. Conversely, AAVs moved unhindered on tyrosinated microtubules, and two examples of AAV motility on tyrosinated microtubules are shown in supplementary video 4 and 5.

We have added these annotations to the movies and assigned appropriate titles to them. We have included figure representations of these different events inferred from the movies in Figure S2H-J. Also, we have added text in the result section to explain the motility behavior (line number 206-210; Video 2- AAV2 localized on detyro-MT shows long pauses, video 3- AAV2 moves unhindered when not on detyro-MT and video 5- AAV2 moves unhindered on tyro-MT).

Minor points:

1) For better readability, all plots and graphs in the figures should carry a short descriptive title above the panels.

We have included a short descriptive title for all the plots and graphs and most image panels in

the revised figure panels.

2) It is not clear in Fig 1D,E how the fraction of detyr- and tyr-microtubules was determined - did the authors classify microtubules into these two categories, and then measure their overall length, or did they simply quantify the fluorescent signal of the staining. As mentioned in the main points, an accompanying western blot would be very helpful here.

We applied intensity thresholding, measured the integrated density of detyrosinated microtubules, and divided this by the corresponding tyrosinated microtubule signal to obtain the ratio. We have corrected the Y-axis labeling in Figure 1D and Figure S1I to reflect this clarity, added a schematic illustrating our quantification strategy in the supplementary results Figure S1A, and cited relevant literature supporting this approach (Shen and Ori-McKenney, 2024).

3) Fig 1J,L,N - it is difficult to see the detyr-tubulin staining in the merged images - the authors should show monochrome panels, at least for the detyr-staining, to allow the reader to get a clear picture of the differences between the represented cells.

We have now revised the figure with separate monochrome and merge panels for clarity.

4) The authors assume that the detyrosinating enzyme responsible for the observed change in tubulin modification is Vash - but did they verify this? There are two Vash enzymes, and two additional enzymes from a different family (MATCAP also known as TMCP1, and the homologue TMCP2; Nicot et al. (2023) A family of carboxypeptidases catalyzing alpha- and beta-tubulin tail processing and deglutamylation. *Sci Adv* 9: eadi7838). Can the authors provide evidence that the Vash enzyme they propose to be in charge of detyrosination after AAV2 infection is indeed Vash1?

As suggested, we tested the effect of both VASH1 and VASH2 on AAV2 transduction and have included these results in the main text (Figure 5A, Figure S5A-D). Our findings indicate that Vasohibins primarily mediate the AAV2-induced detyrosination observed in Huh7 cells. However, we acknowledge that other detyrosination enzymes such as TMCP1/2 may also contribute, particularly in other cell types, and we have highlighted this possibility in the discussion (line number 381-383).

6) Correct bugs in figures: 4D, 4G, 5D, 5E, 5L.

We apologize for the errors and have now corrected them

7) Add cell line name in figure panels if different ones represented in a figure (ex Fig 1D, E).

We have added cell line names to the panels.

References

- Aillaud, C., Bosc, C., Peris, L., Bosson, A., Heemeryck, P., Dijk, J. Van, Le Fric, J., Boulan, B., Vossier, F., Sanman, L. E., et al. (2017). Vasohibins/SVBP are tubulin carboxypeptidases (TCPs) that regulate neuron differentiation. *Science*, **358**(6369), 1448-1453.
- Drabek, K., van Ham, M., Stepanova, T., Draegestein, K., van Horsen, R., Sayas, C. L., Akhmanova, A., ten Hagen, T., Smits, R., Fodde, R., et al. (2006). Role of CLASP2 in Microtubule Stabilization and the Regulation of Persistent Motility. *Current Biology* **16**, 2259-2264.
- Fonrose, X., Ausseil, F., Soleilhac, E., Masson, V., David, B., Pouny, I., Cintrat, J. C., Rousseau,

- B., Barette, C., Massiot, G., et al. (2007). Parthenolide inhibits tubulin carboxypeptidase activity. *Cancer Res* **67**, 3371-3378.
- Genova, M., Grycova, L., Puttrich, V., Magiera, M. M., Lansky, Z., Janke, C. and Braun, M. (2023). Tubulin polyglutamylation differentially regulates microtubule- interacting proteins. *EMBO J* **42**,.
- Ho, K. H., Candat, A., Scarpetta, V., Faucourt, M., Weill, S., Salio, C., D'Este, E., Meschkat, M., Wurm, C. A., Kneussel, M., et al. (2023). Choroid plexuses carry nodal-like cilia that undergo axoneme regression from early adult stage. *Dev Cell* **58**, 2641-2651.e6.
- Hotta, T., Haynes, S. E., Blasius, T. L., Gebbie, M., Eberhardt, E. L., Sept, D., Cianfrocco, M., Verhey, K. J., Nesvizhskii, A. I. and Ohi, R. (2021). Parthenolide Destabilizes Microtubules by Covalently Modifying Tubulin. *Current Biology* **31**, 900-907.e6.
- Li, D., Ding, X., Xie, M., Huang, Z., Han, P., Tian, D. and Xia, L. (2020). CAMSAP2- mediated noncentrosomal microtubule acetylation drives hepatocellular carcinoma metastasis. *Theranostics* **10**, 3749-3766.
- Lu, Y. S., Kashida, Y., Kulp, S. K., Wang, Y. C., Wang, D., Hung, J. H., Tang, M., Lin, Z. Z., Chen, T. J., Cheng, A. L., et al. (2007). Efficacy of a novel histone deacetylase inhibitor in murine models of hepatocellular carcinoma. *Hepatology* **46**, 1119-1130.
- McKenney, R. J., Huynh, W., Vale, R. D. and Sirajuddin, M. (2016). Tyrosination of α -tubulin controls the initiation of processive dynein-dynactin motility. *EMBO J* **35**, 1175-1185.
- Mitra, S., Shanmugapriya, S., Santos da Silva, E. and Naghavi, M. H. (2020). HIV-1 Exploits CLASP2 To Induce Microtubule Stabilization and Facilitate Virus Trafficking to the Nucleus. *J Virol* **94**,.
- Mohan, N., Sorokina, E. M., Verdeny, I. V., Alvarez, A. S. and Lakadamyali, M. (2019). Detyrosinated microtubules spatially constrain lysosomes facilitating lysosome-autophagosome fusion. *Journal of Cell Biology* **218**, 632-643.
- Pietsch, N., Chen, C. Y., Kupsch, S., Bacmeister, L., Geertz, B., Herrera-Rivero, M., Siebels, B., Voß, H., Krämer, E., Braren, I., et al. (2024). Chronic Activation of Tubulin Tyrosination Improves Heart Function. *Circ Res* **135**, 910-932.
- Ramirez-Rios, S., Choi, S. R., Sanyal, C., Blum, T. B., Bosc, C., Krichen, F., Denarier, E., Soleilhac, J. M., Blot, B., Janke, C., et al. (2023). VASH1-SVBP and VASH2-SVBP generate different detyrosination profiles on microtubules. *Journal of Cell Biology* **222**,.
- Sabo, Y., Walsh, D., Barry, D. S., Tinaztepe, S., De Los Santos, K., Goff, S. P., Gundersen, G. G. and Naghavi, M. H. (2013). HIV-1 induces the formation of stable microtubules to enhance early infection. *Cell Host Microbe* **14**, 535-546.
- Shen, Y. and Ori-McKenney, K. M. (2024). Microtubule-associated protein MAP7 promotes tubulin posttranslational modifications and cargo transport to enable osmotic adaptation. *Dev Cell* **59**, 1553-1570.e7.
- Tang, Q., Sensale, S., Bond, C., Xing, J., Qiao, A., Hugelier, S., Arab, A., Arya, G. and Lakadamyali, M. (2023). Interplay between stochastic enzyme activity and microtubule stability drives detyrosination enrichment on microtubule subsets. *Current Biology* **33**, 5169-5184.e8.

Second decision letter

MS ID#: jcs.264190R1

MS TITLE: Suppressing microtubule detyrosination augments AAV2 endosomal escape and gene delivery.

AUTHORS: Shefali Tripathi; Shamshul Huda; Joydipta Kar; Dinesh Chandra; Giridhara R Jayandharan; Nitin Mohan

ARTICLE TYPE: Research Article

Dear Dr Mohan,

We have now reached a decision on the above manuscript.

To see the reviewers' reports and a copy of this decision letter, please go to:

As you will see, the reviewers gave favourable reports but reviewer 2 raised a strong point that I feel needs to be addresses as it will strengthen the papers message. I hope that you will be able to carry these out because I would like to be able to accept your paper when it is returned with the additional data.

Reviewer 1

Advance summary and potential significance to field

The authors have addressed my concerns. Additional controls have been provided and some limitations of the work and potential caveats have been better acknowledged. I am happy to recommend publication.

Reviewer 2

Advance summary and potential significance to field

The authors have addressed most of this reviewer's concerns - this is most appreciated.

One question which remains unanswered though is: what is the effect of enzymatically dead enzymes on AAV2 behaviour in cells? Indeed, when exploring the effect of an enzyme's overexpression (in this case VASH1/2 and TTL) on a biological phenomenon, it is crucial to verify how much the enzymatic activity of the enzyme contributes to the phenomenon, as compared to the overexpression itself. The use of an enzymatically dead enzyme mutant, but not of GFP alone, allows to answer this question. While the equivalent experiments with scrambled shRNA were performed for enzyme knockdown, no such controls have been performed for overexpression.

This reviewer believes that this is an important control to confirm the alleged specific effect of the VASH1/2 and TTL enzymatic activities on AAV2 transport.

Second revision

Author response to reviewers' comments

Dear Editor,

Please find the response to the reviewer's comment below.

Response to reviewer 2

Minor comment

One question which remains unanswered though is: what is the effect of enzymatically dead enzymes on AAV2 behaviour in cells? Indeed, when exploring the effect of an enzyme's overexpression (in this case VASH1/2 and TTL) on a biological phenomenon, it is crucial to verify how much the enzymatic activity of the enzyme contributes to the phenomenon, as compared to the overexpression itself. The use of an enzymatically dead enzyme mutant, but not of GFP alone, allows to answer this question. While the equivalent experiments with scrambled shRNA were performed for enzyme knockdown, no such controls have been performed for overexpression.

In response to Reviewer 2's concern regarding the inclusion of dead-mutant control experiments to complement our overexpression data, we have now incorporated additional results using the catalytically inactive (dead mutant) form of VASH2. This VASH2 dead mutant, is mutated at a cysteine residue to alanine (C158A) and is incapable of catalyzing the microtubule detyrosination (Aillaud et al., 2017). Our new data show that overexpression of the VASH2 dead mutant results in ~ 57% of cells being transduced by AAV2 (Figure A), comparable to the transduction efficiency observed under untreated control conditions (Figure B). Congruently, VASH2 overexpression and knockdown, respectively, decreased and increased AAV2 transduction relative to the catalytically inactive VASH2 mutant (Figure A). These results now provide the appropriate experimental controls requested by Reviewer 2. We have added the new observation in results (line number 271-274), in Figures 5A, B and Figure S5A, their respective figure legends and in the methods section (line 426)

Figure A: Assessment of the impact of VASH and TTL overexpression and VASH shRNA on AAV2 transduction efficiency. Confocal imaging of Huh7 cells, expressing VASH C158A (dead mutant), overexpressing VASH2, expressing VASH shRNA or TTL overexpression (magenta) infected with AAV2 EGFP, where expression of the EGFP transgene (green) indicates AAV2 transduction (A). Percentage of EGFP-positive cells among VASH C158A, VASH2 overexpression, VASH shRNA or TTL-overexpressing populations, highlighting their influence on transduction. n=100 cells analyzed for each condition from triplicate

Figure B: Transduction of AAV2-EGFP (green) in Huh7 cells (nuclei marked with Topro: magenta), in untreated condition (no plasmids and drugs), shows ~55% transduction efficiency.

Reference

Aillaud, C., Bosc, C., Peris, L., Bosson, A., Heemeryck, P., Dijk, J. Van, Le Friec, J., Boulan, B., Vossier, F., Sanman, L. E., et al. (2017). Vasohibins/SVBP are tubulin carboxypeptidases (TCPs) that regulate neuron differentiation. *Science*, 358(6369), 1448-1453.

Third decision letter

MS ID#: jcs.264190R2

MS Title: Suppressing microtubule detyrosination augments AAV2 endosomal escape and gene delivery.

Authors: Shefali Tripathi; Shamshul Huda; Joydipta Kar; Dinesh Chandra; Giridhara R Jayandharan; Nitin Mohan

Article Type: Research Article

Dear Dr Mohan,

I am happy to tell you that your manuscript has been accepted for publication in Journal of Cell Science, pending standard publication integrity checks.